# Contrasting impacts of forests on cloud cover based on satellite observations

Ru Xu[1,2], Yan Li [1,2✉], Adriaan J. Teuling [3], Lei Zhao [4,5], Dominick V. Spracklen [6], Luis Garcia-Carreras [7], Ronny Meier [8,9], Liang Chen [10], Youtong Zheng [11,12], Huiqing Lin [1,2] & Bojie Fu [1,13]

Forests play a pivotal role in regulating climate and sustaining the hydrological cycle. The biophysical impacts of forests on clouds, however, remain unclear. Here, we use satellite data to show that forests in different regions have opposite effects on summer cloud cover. We find enhanced clouds over most temperate and boreal forests but inhibited clouds over Amazon, Central Africa, and Southeast US. The spatial variation in the sign of cloud effects is driven by sensible heating, where cloud enhancement is more likely to occur over forests with larger sensible heat, and cloud inhibition over forests with smaller sensible heat. Ongoing forest cover loss has led to cloud increase over forest loss hotspots in the Amazon (+0.78%), Indonesia (+1.19%), and Southeast US (+ 0.09%), but cloud reduction in East Siberia (-0.20%) from 2002-2018. Our data-driven assessment improves mechanistic understanding of forest-cloud interactions, which remain uncertain in Earth system models.

[1] State Key Laboratory of Earth Surface Processes and Resources Ecology, Beijing Normal University, Beijing 100875, China. [2] Institute of Land Surface System and Sustainable Development, Faculty of Geographical Science, Beijing Normal University, Beijing 100875, China. [3] Hydrology and Quantitative Water Management Group, Wageningen University and Research, Wageningen, The Netherlands. [4] Department of Civil & Environmental Engineering, University of Illinois at Urbana-Champaign, Urbana, IL 61801, USA. [5] National Center for Supercomputing Applications, University of Illinois at Urbana-Champaign, Urbana, IL 61801, USA. [6] School of Earth and Environment, University of Leeds, Leeds LS2 9JT, UK. [7] Centre for Atmospheric Science, Department of Earth and Environmental Sciences, University of Manchester, Manchester M139PL, United Kingdom. [8] Institute for Atmospheric and Climate Science, ETH Zurich, 8092 Zurich, Switzerland. [9] Umweltschutz, Stadt Luzern, 6005 Luzern, Switzerland. [10] Climate and Atmospheric Sciences Section, Illinois State Water Survey, Prairie Research Institute, University of Illinois at Urbana-Champaign, Champaign, IL 61820, USA. [11] Program in Atmospheric and Oceanic Sciences, Princeton University, Princeton, NJ 08544, USA. [12] NOAA/Geophysical Fluid Dynamics Lab, Princeton, NJ 08544, USA. [13] State Key Laboratory of Urban and Regional Ecology, Research Center for Eco-Environmental Sciences, Chinese Academy of Sciences, Beijing 100085, China. ✉email: yanli.geo@gmail.com

F orests regulate climate and sustain the hydrological cycle through biophysical processes[1,2]. These processes are tightly linked to land surface properties, such as albedo, roughness, and canopy conductance that affect the exchange of energy and water between the land and the atmosphere[1,2]. The direct bio-physical impacts of forests on surface temperature have been extensively studied, revealing a latitudinal transition from tropical cooling to boreal warming[3–5]. However, less attention has been paid to its indirect impacts on clouds and precipitation, two physically linked key components in the hydrological cycle. How clouds and precipitation respond to land cover change has been poorly constrained and presents one of the major challenges in climate change assessment[6].

Global climate models (GCMs) have predicted a reduction in precipitation and a frequent decrease in cloud cover resulting from large-scale deforestation, with the greatest decrease in tropical regions[7–9]. Although these results generally support that vegetation enhances clouds and precipitation at large-scales[10,11], especially in the tropics, these continental- or global-scale land clearing experiments implemented in models with a relatively coarse resolution are not consistent with the ongoing small-scale land activities in the real world. Results from these GCM experiments are often complicated by mixing the local-scale intrinsic biophysical mechanism[3] with the nonlocal feedbacks triggered by the large-scale land cover change in the climate system, making it hardly comparable with observations[12,13].

In contrast to cloud and precipitation reduction simulated in the GCM experiments[7,8,14], high-resolution regional climate models[15,16] and empirical analyses using satellite imagery[17,18] reported that small-scale deforestation increases rather than decreases clouds and precipitation in the Amazon due to land surface heterogeneity[19]. These results revealed inhibited clouds over some forests (e.g., West Africa[20]) at a realistic scale which seemingly contradicts the highly hypothetical GCM results[21] and the enhanced cloud observations over forests in other regions (e.g., western Europe[22] and Central America[23]).

These inconsistent findings among modeling and observational studies highlight the large uncertainty in cloud and convection representations in climate models[24,25] as well as the complexity of forest–cloud interactions, which involve different mechanisms across different scales with varying regional importance[26]. The global pattern of forest impacts on cloud cover, especially how it is shaped by the interplay of different mechanisms, remain largely unresolved.

In this study, we use multi-source satellite observations of high spatial resolution and long-term global coverage to assess the cloud effects of forests across the globe. Furthermore, we explore the potential mechanisms and quantify the cloud effects of forest loss in the past two decades. Our results reveal the contrasting cloud effects of global forests due to moist convection and mesoscale circulation processes, in which sensible heat plays a critical role in differentiating the sign of cloud effect. We find the emergence of ongoing forest loss as an important driver for local cloud cover change, leading to opposite changes in different regions.

## Results

**Potential effects of forests on cloud cover.** Using a space-for-time approach, we define the potential cloud effect of forests as the multiyear mean cloud difference between unchanged forests and nearby non-forest pixels ($\Delta$Cloud = Cloud$_{forest}$ − Cloud$_{nonforest}$). Here, the cloud effect is measured by cloud cover fraction derived from Moderate Resolution Imaging Spectroradiometer (MODIS) and Meteosat Second Generation (MSG) satellite data, which represent the occurrence frequency of clouds over a period of time with a valid range of 0–1 (Supplementary Fig. 1). The positive and

negative $\Delta$Cloud denote enhanced and inhibited cloud cover over forests, respectively. $\Delta$Cloud is estimated globally through a moving window sized at $0.45 \times 0.45°$ ($9 \times 9$ cells) near locations that underwent forest cover change during the study period (see Methods). This approach can minimize cloud effects resulting from large-scale circulation/climate changes, which affect both forests and non-forest. The climatological approach also effectively removes stochastic cloud differences between forests and non-forest caused by individual meteorological events and wind direction changes. Here we first focus on boreal summer months (JJA), during which we expect to observe the most pronounced cloud differences between forests and non-forest[22] for the majority of the northern hemisphere, while results for other seasons are presented later.

Forests exhibit regionally varying effects on JJA cloud cover based on MODIS data (overpass at 13:30 local time, Fig. 1a). Most temperate and boreal forests in Eurasia and North America have higher cloud fractions than non-forest, indicating a cloud enhancement effect (positive $\Delta$Cloud) accounting for 63.21% of all grid samples with a global mean magnitude of +0.0133. In contrast, forests in South Amazon, Central Africa, and Southeast US have lower cloud fractions than nearby non-forest, signifying a cloud inhibition effect (negative $\Delta$Cloud) over the forest with a global mean magnitude of −0.0115. The strength of these contrasting cloud effects (i.e., cloud enhancement and inhibition) follows a latitudinal dependency with the largest magnitude in the tropical regions and diminished toward higher latitudes (Fig. 1b). This is likely due to preferential conditions for convection development at low latitudes, as indicated by their high convective available potential energy, which decreases at higher latitudes[27]. Our additional sensitivity tests indicate that the global pattern of $\Delta$Cloud holds when estimated using alternative window sizes (Supplementary Fig. 2) and split time periods (2002–2007, 2008–2013, 2014–2018, Supplementary Fig. 3), suggesting the robustness of results to scale of a local window and interannual variability of cloud cover.

Similar spatial and latitudinal patterns can also be seen from geostationary MSG satellite data with high temporal resolution (i.e., hourly) but non-global coverage. At 14:00 local time, cloud inhibition is stronger in Central Africa while weaker in the Amazon regions compared to MODIS data (Fig. 1c, d). The hourly MSG cloud data reveal a pronounced diurnal cycle in the cloud effects (Fig. 1e and Supplementary Fig. 4). Consistent with the daytime prevalence of convection, the maximum effect during the entire day (the largest $\Delta$Cloud regardless of sign) occurs mostly during daytime (6 a.m. to 18 p.m. 70%), especially during the afternoon (12 to 18 p.m., 48%) in tropical regions.

The estimated cloud effects of forests could be confounded by orographic clouds because of the dual influences of topography on forest distribution and cloud formation. Human impacts resulted in a global tendency of existing forests to be located at more complex terrain with a higher elevation than non-forest[28]. The high elevation per se could facilitate cloud formation through the orographic lifting of moist air[29], leading to increased cloud cover over the forest located at a higher altitude (Supplementary Fig. 5). To isolate the orographic cloud effect, we decompose $\Delta$Cloud into contributions of tree cover and elevation (Supplementary Fig. 6). We find that the global pattern of $\Delta$Cloud is dominated by tree cover induced cloud effects (41% grid boxes for cloud enhancement and 22% for cloud inhibition), followed by elevation-induced cloud effects (30%), and unexplained effects due to other factors (7%) (Fig. 1f). This confirms that most observed cloud effects are robust features attributable to tree cover rather than topography and other factors.

The MODIS and MSG cloud cover data provide a combined measure of cloud cover fraction, but they do not separate different

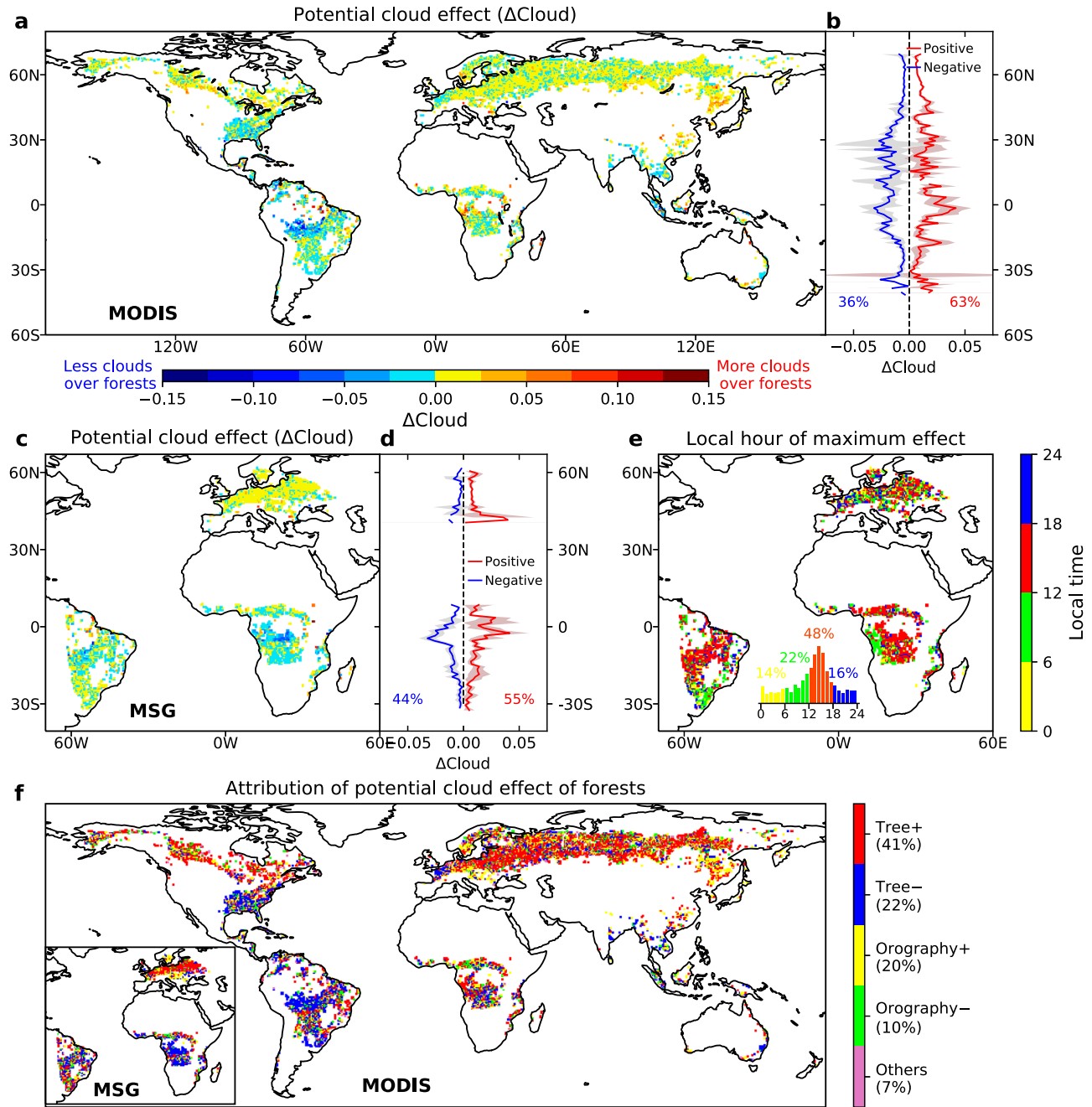

**Fig. 1 The potential effects of forests on June-August (JJA) cloud cover fraction and their attribution.** The potential effect is defined as the differences in cloud cover fraction between forests and nearby non-forest (ΔCloud) from MODIS and MSG satellites that detect clouds. **a** Potential effects of forests on cloud cover fraction based on MODIS data from 2002 to 2018 (overpass at 13:30 local time) and **b** their latitudinal patterns with cloud enhancement and inhibition effects separated. **c**, **d** Potential effects of forests on cloud cover fraction based on hourly MSG data from 2004 to 2013 (overpass at 14:00 local time) and **e** the timing of the maximum effect during a day. The numbers in panels **b** and **d** show the percentage of cloud enhancement (red) and inhibition (blue). **f** Attribution of cloud effects of forests to tree cover and elevation based on MODIS and MSG data. The five attribution categories include tree cover induced cloud increase (Tree+) and decrease (Tree−), orography induced cloud increase (Orography+) and decrease (Orography−), and other unexplained effects. The percentage of each attribution category is calculated based on the MODIS results.

cloud types. By utilizing Sentinel-5P cloud data and a cloud classification scheme[30], we are able to estimate cloud effects of forests with respect to different cloud types (see Methods). We find that globally, cloud effects are dominated by convective clouds in 45.01% of grid boxes, largely contributed by shallow convective stratocumulus clouds (39.10%) (Supplementary Fig. 7). Regionally, the convection dominance becomes more prominent, contributing to 70.43% of cloud effects in the Amazon region.

These further confirm that the cloud effects of forests studied here are primarily convection-driven, as also implied by MODIS and MSG results.

In terms of seasonality, there are notable and region-specific variations in ΔCloud from both MODIS and MSG data (Fig. 2 and Supplementary Fig. 8). The maximum cloud effects appear in local summer for most areas of the northern hemisphere (JJA, June to August) and the mid-latitudes of the southern hemisphere

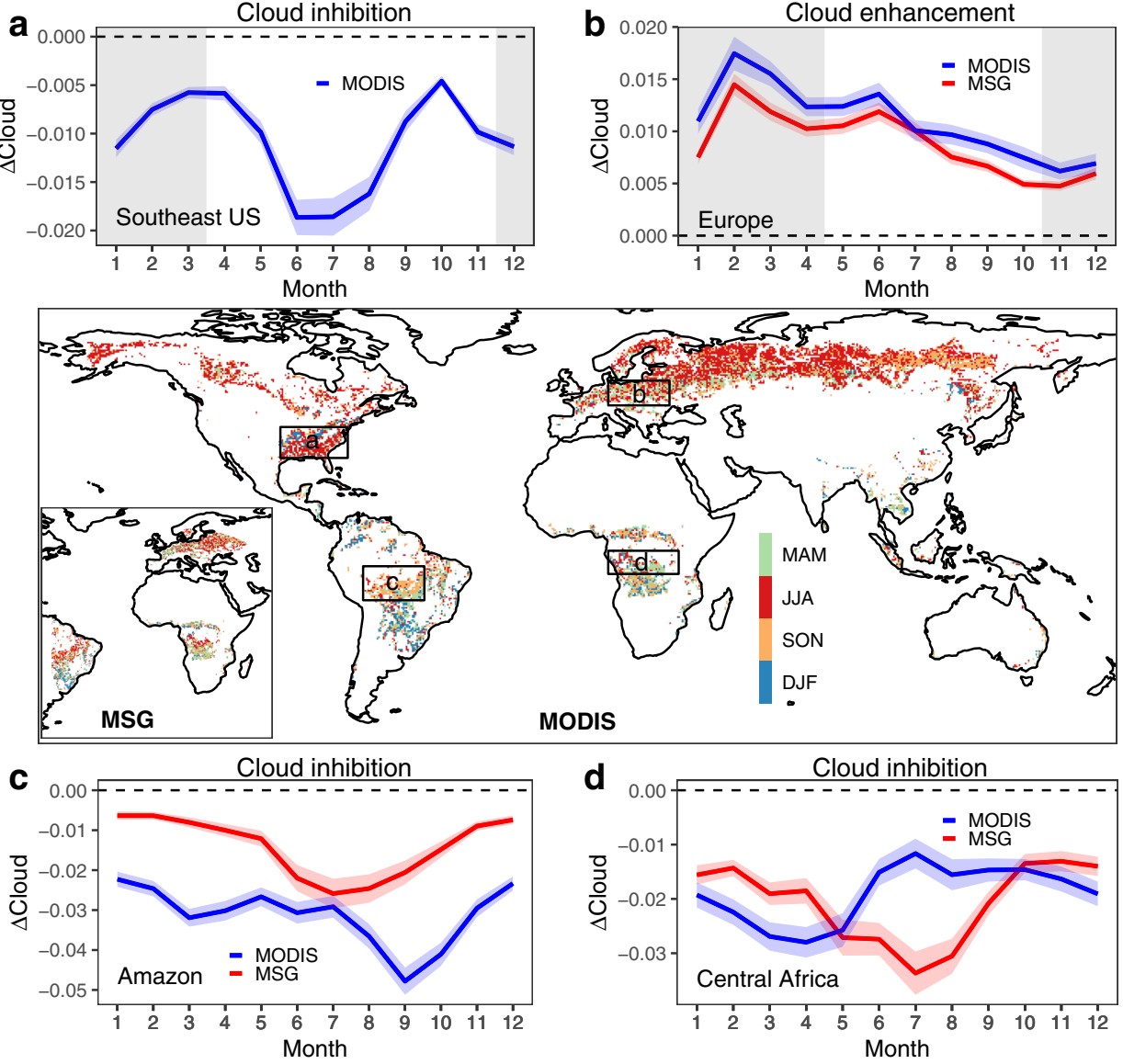

**Fig. 2 Seasonal variations of the potential effects of forests on cloud cover.** Months with the maximum magnitude of ΔCloud for MODIS and MSG during the snow-free season are shown on the global map. MAM is March-April-May, JJA is June-July-August, SON is September-October-November, and DJF is December-January-February. Seasonal variations of ΔCloud are shown in selected regions: **a** Southeast US (97°W to 75°W, 30°N to 40°N), **b** Europe (10°E to 30°E, 47°N to 55°N), **c** Amazon (70°W to 50°W, 16°S to 5°S), and **d** Central Africa (10°E to 33°E, 7.5°S to 0°) for MODIS and MSG. Months with snow cover are shown as the gray shaded areas in the background. The blue and red shaded areas denote the 95% confidence interval of ΔCloud for MODIS and MSG.

(DJF, December to February). In the tropics, this occurs during the dry-wet transition, consistent with existing evidence[16,18,31,32], cloud inhibition in the Amazon is stronger during the dry season (May to November) than in the wet season, although their timings of maximum effect differ (September for MODIS and July for MSG). Interestingly, cloud inhibition in Central Africa exhibits a larger effect during the dry season in MSG data (predominant JJA) but during the wet season in MODIS data (mixed JJA and MAM). In temperate regions, cloud inhibition in the Southeast US is larger in summer, while cloud enhancement in Europe shows a slight decline during the snow-free period.

**The mechanisms of contrasting cloud effects of forests.** While different biophysical processes are involved in the forest–cloud interactions, it has been unclear which factors determine the spatial occurrences of cloud enhancement and inhibition over

different forests. The geographic variations in specific land cover types of the global forest and non-forest vegetation types show little spatial resemblance to ΔCloud (Supplementary Fig. 9). In terms of biophysical differences, forests generally have reduced albedo, higher roughness, lower land surface temperature (LST), increased evapotranspiration, and soil moisture compared to non-forest vegetation[4,5]. However, these differences are common to almost all forests and cannot explain the contrasting cloud effects, as indicated by their mismatched spatial patterns with ΔCloud (Supplementary Fig. 10).

We find that the sensible heat difference between forest and non-forest (ΔH) is an effective differentiator for the sign of cloud effect among other land surface properties[33]. This is obtained by analyzing the relationship between ΔCloud and ΔH derived from three independent datasets based on satellite[4], a simulation of the Community Land Model (CLM) version 5 (ref. [34]), and 28 paired

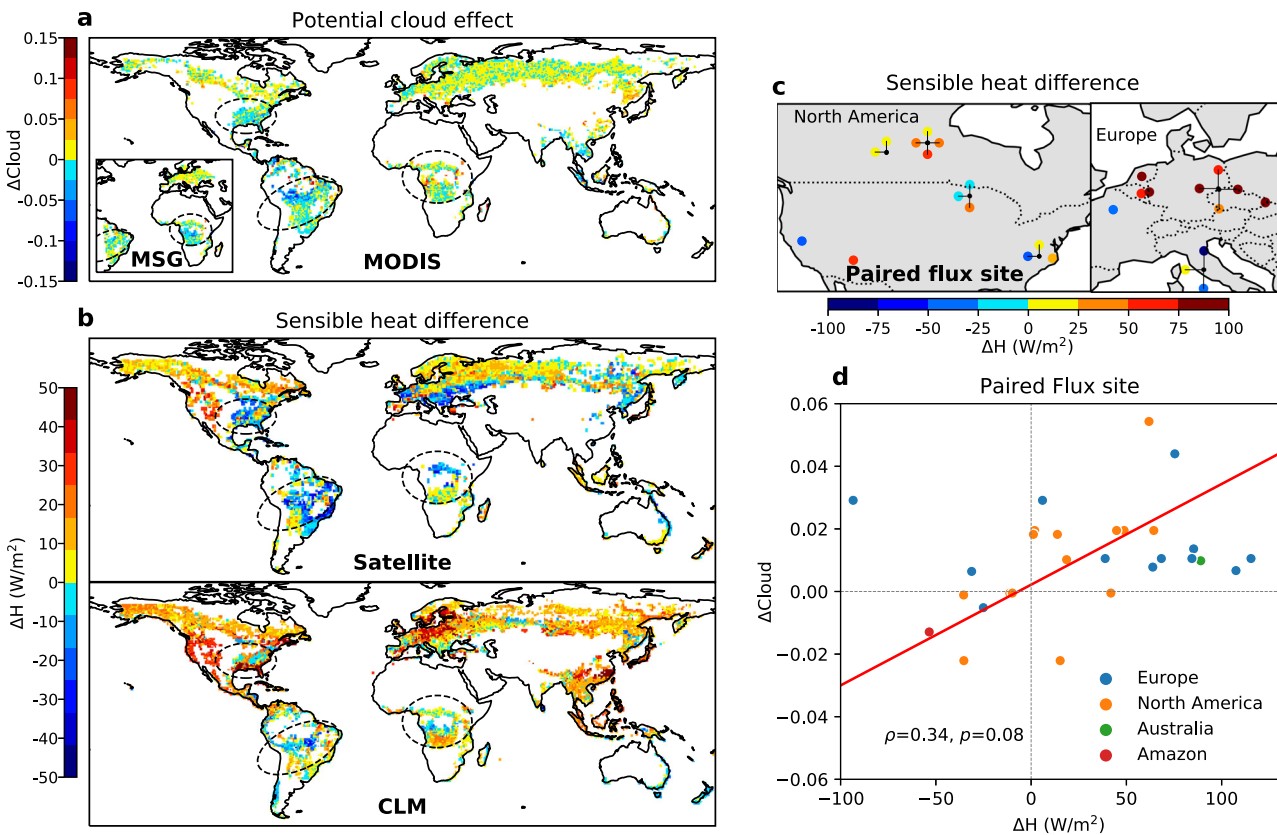

**Fig. 3 The sensible heat difference between forests and non-forest and its relationship with cloud effects. a** Potential effects of forests on cloud cover from MODIS and MSG data (duplicates from Fig. 1a, b). **b** The sensible heat differences between forests and non-forest (ΔH) estimated from satellite data[4], Community Land Model (CLM), and **c** Twenty-eight paired forest and non-forest flux sites (including one Amazon pair not shown on the map, see Supplementary Table 1). The three circles marked in **a** and **b** denote the locations of cloud inhibition which correspond to the negative ΔH in Amazon, Central Africa, and Southeast US. The connection lines with a dot in panel **c** indicate the location of flux tower clusters where multiple flux pairs are close in distance. **d** The relationship between sensible heat (ΔH) and cloud differences between forest and non-forest (ΔCloud) at paired flux towers. The cloud effects at paired flux site locations are extracted from ΔCloud aggregated to 1° based on MODIS data. The line is fitted by geometric mean regression[80]. The Spearman's correlation coefficient (ρ), which is a nonparametric measure of rank correlation, is calculated by the *spearmanr* function of scipy.stats module in Python with its *p* value (*p*) determined by a two-tail *t*-test.

forest and non-forest flux sites[35] (Fig. 3a–c). Both the satellite and the CLM results indicate that cloud inhibition (negative ΔCloud) mainly occurs at locations where forests exhibit a smaller sensible heat flux than non-forest (negative ΔH), including southern Amazon[36], Central Africa, and the Southeast US (marked by three circles in Fig. 3a, b). By contrast, cloud enhancement (positive ΔCloud) in the rest of the world broadly corresponds to locations with higher sensible heating over forests (positive ΔH), despite few inconsistencies in southern Europe among the considered datasets. Such a spatial co-occurrence is further confirmed by the positive relationship between observed ΔH from paired flux sites and MODIS ΔCloud (ρ = 0.34, Fig. 3d), and those between ΔH derived from satellite/CLM and MODIS ΔCloud (weaker but statistically significant), suggesting that cloud enhancement is more likely to occur when sensible heat over the forest is larger than nearby non-forest, and cloud inhibition occurs when sensible heat over the forest is smaller. Further evidence comes from closely tracked seasonality between ΔH and ΔCloud at paired flux sites (Supplementary Fig. 11). The larger sensible heat over forests corresponds to greater cloud enhancement (e.g., Europe), whereas the lower sensible heat over forests correspond to stronger cloud inhibition (e.g., Southeast US and Amazon).

The spatial patterns of ΔH reflect the biophysical and climatic controls on energy redistribution in forest and non-forest along with latitude and moisture levels[37,38]. Forests at low latitudes

under humid climates have smaller Bowen ratios as most available energy goes into latent heat rather than sensible heat, resulting in even smaller sensible heat compared to non-forest. In comparison, forests at higher latitudes under drier climates have larger Bowen ratios leading to the opposite effect. The collective evidence strongly suggests a central role of sensible heat in convection triggering and cloud formation[33]. A higher sensible heat relative to nearby landscape is indicative of a preferable condition for convection and cloud development, though it is initiated by different mechanisms for the enhanced and inhibited cloud cover over forests.

The mechanisms of enhanced cloud over forests are associated with several interconnected processes operating at different scales that are conducive to the growth of moist convection (Fig. 4a). Compared with non-forest vegetation, forests usually exhibit high evapotranspiration[5], which provides abundant water vapor supply for cloud formation and sustains moisture recycling over large-scales[39,40]. The low albedo and high roughness of forests promote a greater fraction of incoming solar energy partitioned into turbulent heat fluxes, increasing turbulent mixing and convective instability in the boundary layer[16,32,41]. At small scales, the differential roughness between forest and non-forest induces frictional convergence in downwind direction[22,42]. Enhanced sensible heating, which typically occurs over the forest relative to non-forest vegetation[35], serves as a major lifting

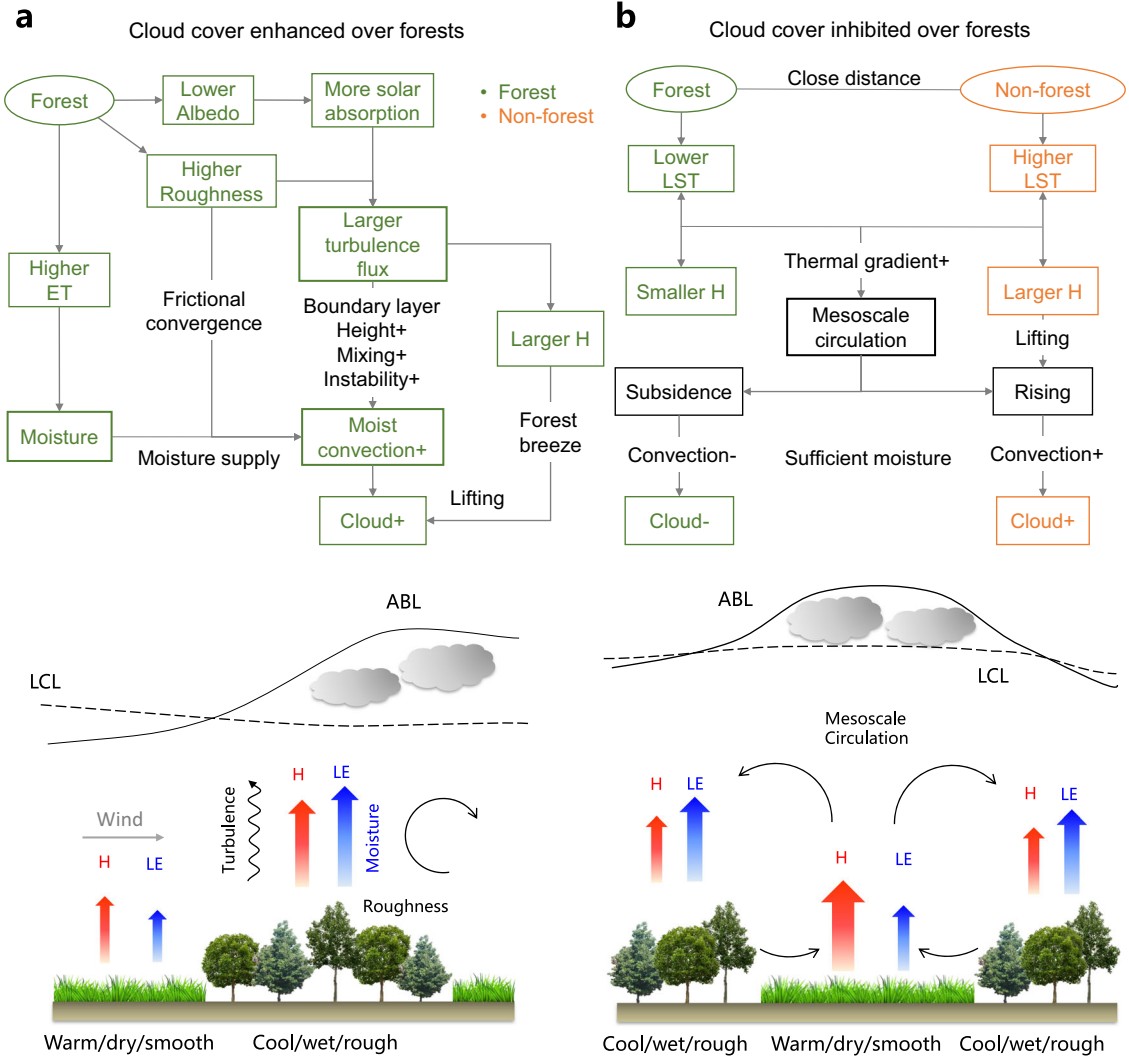

**Fig. 4 Mechanisms of contrasting cloud effects of forests. a** Clouds enhanced over forests through increased convection due to increased moisture supply and turbulence. **b** Clouds inhibited over forests through suppressed convection due to divergence of mesoscale circulations. ABL atmospheric boundary layer, LCL lifting condensation level, LE latent heat, H sensible heat.

mechanism to initiate convection and the growth of boundary layer[32,33], and the formation of a "forest-breeze" analogous to a sea-breeze. Specifically, the high sensible heat elevates the atmospheric boundary layer (ABL) such that the lifting condensation level (LCL) is lower than the ABL depth[18,32,40], thereby supporting low-level cloud formation.

The mechanisms of inhibited cloud cover over the forest and enhanced cloud cover over nearby non-forest, are likely linked to an opposite mesoscale circulation triggered by heat and moisture anomalies of heterogeneous landscape between forests and non-forest[43] (Fig. 4b). Differential heating between forests (cooler, wetter) and non-forest (warmer, drier) creates a thermally-driven mesoscale circulation with downward flow over forests. The rising airflow over non-forest initiates convective clouds while the subsidence branch over forests outweighs moist convection processes and inhibits cloud development[44]. The warmer deforested areas with larger sensible heat flux, combined with increased atmospheric instability[16] can reinforce mesoscale circulation and provide a favorable environment for cloud formation[15,32,45]. Moreover, the cloud effects, once developed, tend to dampen the sensible heat flux differences from which they originate. The inhibited cloud cover (with lower sensible heat) over forests, in turn, enhances sensible heat as the land surface

receives more incoming shortwave radiation. It also implies a memory effect as the cloud effects detected at the satellite overpass time reflect the flux differences accumulated earlier during the morning and noon, while the developed cloud effects dampen the flux differences in the afternoon.

The development of mesoscale circulation depends on the length scale of the land heterogeneity. Mesoscale circulation is sensitive to spatial scale and is typically generated at scales of 10–100 km[16,41]. To investigate the sensitivity of cloud inhibition effect induced by mesoscale circulation to spatial scale, we re-estimated ΔCloud using MODIS cloud data resampled to different spatial resolutions. We find that with reduced resolutions of cloud data, the spatial coverage of cloud inhibition shrinks from ~37% at 0.05° to ~24–28% at 1°, while cloud enhancement becomes more dominant (from 63% to ~76–72%) (Supplementary Fig. 12 and Supplementary Table 2). This implies that at coarser scales (e.g., typical GCM spatial resolutions), at which mesoscale processes become less important (i.e., less cloud inhibition), observation- and model-based results tend to converge on cloud enhancement of forests.

The strength and position of mesoscale circulation, as well as the resultant cloud effect, are influenced by synoptic conditions as well. Mesoscale circulations get intensified under weak synoptic

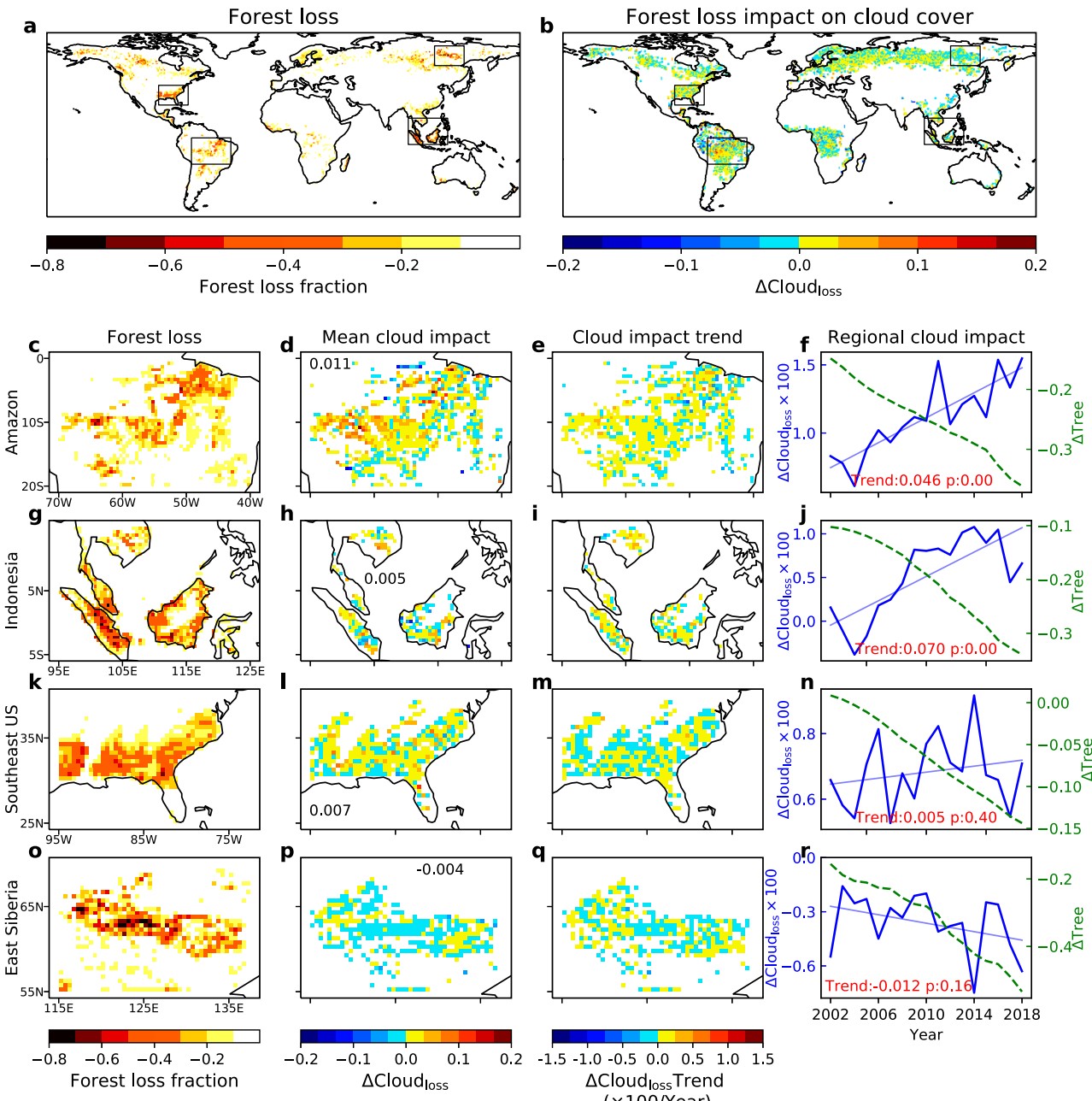

**Fig. 5 Impacts of forest loss on JJA cloud cover based on MODIS data from 2002 to 2018. a** The accumulated forest loss fraction from 2001 to 2018. **b** The actual cloud impact of forest loss ($\Delta Cloud_{loss}$), defined as the mean cloud difference between forest loss location and nearby unchanged forests from 2002 to 2018. Four hotspot regions that experienced intensive forest loss are highlighted in panels **c** to **r**, showing their forest loss fractions, mean $\Delta Cloud_{loss}$ during the study period, and regional and temporal trends of $\Delta Cloud_{loss}$ between 2002 and 2018 (column-wise). The cloud impacts in forest loss hotspot regions are estimated from grid boxes with tree cover loss fraction >0.05. The green dashed line in the last column (**f**, **j**, **n**, **r**) shows the annual tree cover difference between forest loss location and nearby unchanged forests ($\Delta Tree$); the solid blue line shows the temporal trends of $\Delta Cloud_{loss}$ (unit: %/year); and the thin blue line and text show the linear fitted trend and its $p$ value.

conditions, because large wind flow eliminates the thermal gradient generated by land heterogeneity[16]. For example, the cloud inhibition effect in Amazon forests becomes stronger during the dry season when synoptic winds are weaker and the LST gradient is larger[5,15,18]. Under moderate winds conditions, due to low-level advection, the convergence over the forest could be shifted along the wind direction, causing enhanced convection downwind and suppressed convection upwind of the forest[22,45]. The variations in cloud effects due to varying synoptic conditions are most prominent at fine temporal scales for a specific region,

which may not necessarily be manifested in our global pattern derived at climatological scales.

The mesoscale circulation mechanisms associated with the inhibited clouds over forests echo a large body of research in land–atmosphere interactions emphasizing the critical role of surface fluxes and soil moisture anomalies in the atmospheric boundary layer, clouds, and precipitation processes[46,47]. The preference of convective clouds and precipitation over drier soils identified in previous studies using observations[48,49], analytic models[50], and numerical simulations[51] is in line with the

enhanced clouds over regions of higher sensible heat flux, as we show here for forest and non-forest transitions. Interestingly, the preferences of clouds over non-forest found in Southeast US, Amazon, and Central Africa of our study also roughly correspond to those reported to exhibit a temporal preference of afternoon rain with drier soil conditions (Central US, West Amazon, and parts of the Sahel and Equatorial Africa)[49].

**Cloud effects of forest loss over the last two decades.** Forest cover loss has been rapidly occurring globally over the last two decades, especially in tropical regions owing to continuous deforestation (Fig. 5a)[52,53]. These changes are expected to cause different cloud responses in forests with enhanced or inhibited cloud effects. We quantify the actual cloud impact of forest loss that has already occurred by comparing cloud fraction at locations that underwent net tree cover losses with nearby unchanged forests since 2000 (Fig. 5b) in four hotspot regions of forest loss (Fig. 5c, g, k, o).

During the study period, forest loss enhanced cloud cover in three of those hotspot regions. The mean cloud cover fractions at forest-loss locations (tree cover loss >0.05) are on average 0.011, 0.005, and 0.007 higher than nearby unchanged forests in the Amazon, Indonesia, and Southeast US, respectively (Fig. 5d, h, i). Furthermore, cloud enhancement in these forest loss hotspots became increasingly stronger with the decline and fragmentation of tree cover[54], which translates into total cloud fraction increases of 0.78% (0.046%/year, $p < 0.01$), 1.19% (0.070%/year, $p < 0.01$), and 0.09% (0.005%/year, $p = 0.40$) throughout 2002 to 2018 for Amazon, Indonesia, and Southeast US hotspots, respectively (3rd and 4th column in Fig. 5). Note that in the Amazon, forest loss legacy before 2001 had already caused lower tree cover ($\Delta$Tree $< -0.1$) in 2002, which is responsible for the enhanced cloud cover (positive $\Delta$Cloud$_{loss}$) observed over the forest loss locations at the beginning of the study period (Fig. 5f). However, enhanced clouds over deforested regions require the retaining of nearby forest patches over which clouds are reduced. As the scale of deforestation increases with fewer forest patches left, the mesoscale circulation induced cloud enhancement over deforested locations would decrease and ultimately transit to a cloud reduction regime[14,45,55,56]. Unlike other hotspots, East Siberia is a region where forest loss induced cloud cover reduction. The mean cloud cover is 0.004 lower than nearby unchanged forests over the forest loss location (Fig. 5p). The cloud reduction also exhibited a strengthening trend, resulting in a total reduction in cloud cover fraction of $-0.20\%$ ($-0.012\%$/year, $p = 0.16$) from 2002 to 2018 (Fig. 5q, r). These results provide strong evidence that ongoing forest loss could emerge as an important driver for local cloud cover change, especially over areas with intensive forest loss[56,57]. However, due to the decreasing magnitude of cloud effect and increasing cloud interannual variability toward higher latitudes (Fig. 1 and Supplementary Fig. 1), the signal-to-noise ratio of the cloud effect also decreases at higher latitudes, making forest loss induced cloud cover changes there less detectable than those occurring at lower latitudes.

## Discussion
This study offers global-scale observational evidence for contrasting cloud effects of forests and advances our mechanistic understanding of forest–cloud interactions. The cloud effects estimated in our study reflect the local impact of forests on cloud cover and is, therefore, more representative of real-world small-scale forest cover change, without generating the large-scale climate feedbacks which are usually triggered in GCM experiments[3,12]. The local perspective allows us to identify the role of mesoscale circulation which is limited to small scales, a

feature that cannot be resolved by global climate models and is likely the cause of the discrepancy in clouds and precipitation responses between climate model and observational studies, as also shown for soil moisture[25]. Although cloud processes are far more complicated than reflected in the cloud cover observations, our analysis provides a first-order approximation and benchmark for the forest–cloud interactions at fine scales. These results can help constrain convection and cloud processes in climate models which are often parameterized and subject to large uncertainty.

It is worth noting that the estimated cloud effects, despite their broad agreement in the global pattern across datasets, can differ in magnitude and even in sign in certain regions (e.g., the inconsistent cloud effects and their seasonality in Central Africa), suggesting cloud data are a key uncertainty source for our analysis. The cloud effects of forests estimated in a recent study[58] based on a different MODIS-derived cloud dataset[59] (produced by a different retrieval algorithm) revealed a similar global dominance of cloud enhancement and regional prevalence of cloud inhibition (e.g., in Amazon). Yet regional inconsistencies remained for Southeast US and Central Africa, where the identified cloud inhibition could be weakened or absent with alternative cloud data (ref. [58] and Supplementary Fig. 13). This emphasizes the need of using multi-source cloud data to improve the robustness of the estimated cloud effects while reducing the uncertainty from data. Nevertheless, the occurrence of cloud inhibition in these regions is indirectly supported by lower sensible heat over forests, which is in line with the mesoscale circulation mechanism, as well as the consistency between potential cloud inhibition and the actual cloud increase over forest loss locations (i.e., Southeast US hotspot). While more direct observational evidence is always desirable to help resolve inconsistencies, the lack of observations in regions like Central Africa, which have received less attention than the Amazon, hinders comparisons against other available evidence. This highlights the importance of dedicated observational efforts in specific regions, especially those understudied, to provide complementary information to our global-scale analysis.

Given the tight coupling of cloud and precipitation processes, the cloud impact of forest cover change may translate into precipitation impact[60]. Observational evidence exists in the Amazon where the cloud increase in deforested areas has been accompanied by a precipitation increase[61,62]. Although it is hard to directly detect the precipitation impact of deforestation from observations[8], the cloud impact derived from high-resolution satellite data could provide helpful inference to potential precipitation change, especially in tropical regions where convective rainfall is dominant[63]. However, the distinct roles of different cloud types (e.g., shallow cumulus clouds or deep convective clouds) in precipitation and radiative processes complicate the inference from clouds to precipitation changes. Therefore, the extent to which forest loss induced cloud change translates to precipitation may depend on their regional-, seasonal-, and cloud type-specific interactions and require further investigation.

Our results show ongoing forest cover loss has become an important driver of local cloud change over areas with intensive forest loss, which could potentially modify precipitation patterns[56,57,64] and in turn, impose additional feedbacks to (either amplify or dampen) temperature change through clouds' radiative effects[58,65]. Retaining forest patches could enhance cloud cover over nearby agricultural lands through mesoscale circulation (e.g., in the Amazon)—with positive benefits of reduced temperature and possibly increased rainfall. Conversely, the reduction in cloud cover over remaining forest patches may reduce the resilience of the forest to future climate change[66]. Moreover, the changing forest cover owing to either deforestation, increased tree vulnerability under future warming[67,68], or

afforestation[58] will not only affect local climate and hydrology, but also cause remote impacts on distant regions through moisture recycling and advection[69] and have other ecological and socioeconomic implications[56,70]. An accurate prediction of these impacts would benefit from an improved understanding of forest–cloud interactions, which could be facilitated by the cooperation of remote sensing of high spatial-temporal resolutions and climate models that can better characterize mesoscale cloud processes.

## Methods

**Cloud cover and environmental datasets.** The monthly mean MODIS cloud fraction at 0.05° used in this study was computed from the daily cloud mask data ("cloudy" label for the bits 0–1 of "state_1 km" band) included in the MODIS Surface Reflectance product (MYD09GA.006, overpass at local time of 13:30) of Aqua from 2002 to 2018, using the reduceResolution function with "mean" aggregation method on Google Earth Engine (https://earthengine.google.com/). The 1-km cloud mask was produced based on the MOD35_L2 cloud mask product, which had been extensively validated[71,72]. Before computing cloud fractions, a snow/ice flag (the bit 12 of "state_1km" band) was used to remove snow or ice pixels in the cloud record because the high reflectivity of snow/ice degrades the accuracy of cloud detection, especially during winter in the northern hemisphere. Therefore, the estimated cloud effect would have larger uncertainty in boreal winter than in summer.

To complement MODIS-based cloud analyses, we used the Meteosat Second Generation (MSG) hourly cloud fraction data of 2004–2013 at a spatial resolution of 0.05°. The Coordinated Universal Time (UTC) of the raw MSG hourly cloud cover data was converted to local time before being used for analysis.

The cloud fraction from Sentinel-5P Near Real-Time (NRTI) data product was used in this analysis. This dataset is available from 2018-07-05 at a spatial resolution of 0.01° and it has an overpass time of 13:30 similar to MODIS. The Sentinel-5P cloud data, although having a short period of 2 years, allows for the separation of cloud effects into different cloud types, with the help of a cloud classification scheme based on cloud top pressure and cloud optical depth information[30].

Environmental variables include evapotranspiration (ET, MOD16A2 V6), land surface temperature (LST, MYD11A1 V6) from MODIS, and soil moisture (SM) from the TerraClimate dataset. All these environmental variables were averaged into monthly means at 0.05° resolution.

Elevation data are from SRTM Digital Elevation Data at 0.05° resolution. Land cover data include MODIS (MOD12C1) and European Space Agency (ESA) global land cover products, which were aggregated to 0.05°.

**Defining forest cover change.** To define forest/non-forest and forest cover change, we used the Global forest cover (GFC) product which provides global tree cover for the year 2000 (baseline), yearly forest loss from 2001 to 2018, and forest gain from 2000–2012 at 30 m resolution[53]. The GFC data were aggregated to fractions at 0.05°. Net forest cover change was calculated as the sum of the loss and gain accumulated throughout the study period. Pixels with net forest cover change fractions smaller than 0.05 are considered to be "unchanged" and greater than 0.05 are considered to be "changed". Unchanged forests and unchanged non-forest were defined as pixels with baseline tree cover fraction greater or less than 0.5 and with net forest change <0.05. For unchanged non-forest, pixels classified as water, snow/ice, or wetland were excluded using the major composite of MODIS land cover from 2002 to 2005 with the International Geosphere-Biosphere Program (IGBP) classification scheme. For "changed" forest pixels, forest loss was identified as those with a net forest loss >0.15. Forest loss defined this way is expected to pose a stronger signal on clouds than that with a lower threshold, and thus improves the detectability of cloud impact against natural variability of cloud cover.

**Estimating potential and actual impacts of forest loss on cloud cover.** The potential effect of forest on cloud (ΔCloud) was quantified as the mean cloud difference between unchanged forests and nearby non-forest as:

$$\Delta\text{Cloud} = \text{Cloud}_{\text{forest}} - \text{Cloud}_{\text{nonforest}} \tag{1}$$

where $\text{Cloud}_{\text{forest}}$ and $\text{Cloud}_{\text{nonforest}}$ are multiyear or yearly mean cloud fractions averaged over unchanged forest and unchanged non-forest pixels, respectively. ΔCloud defined this way, with the reversed sign, represents the potential impact of forest loss on cloud cover at a given location. The methodology is designed to isolate the cloud effects of land surface conditions from those caused by meteorological conditions. It refers to local cloud impact (caused by land surface conditions) because effects from synoptic conditions and large-scale circulation changes/climate changes (meteorological conditions) are shared by both forest and non-forest and are therefore minimized through subtraction. If there is no effect of forests on cloud cover, the resulting ΔCloud would show random patterns with mixed positive and negative values instead of a systematic pattern, which indicates a cloud preference over forests or non-forest.

To implement Eq. 1, we used a moving window approach to search for comparison samples between forest and nearby non-forest pixels at locations that underwent "forest change" (i.e., net forest change >0.05) across the globe[73]. Each moving window was sized at 9 × 9 pixels (0.45° × 0.45°) and two adjacent windows were half-overlapped with a distance of 5 pixels (i.e., the centers of two windows were 5 pixels apart along latitudinal and longitudinal direction). To avoid cloud inhibition effects from water bodies[74], water pixels and their one-pixel buffer zone were masked out in the window searching strategy for ΔCloud. Therefore, ΔCloud can be calculated using unchanged forest and non-forest pixels within each moving window. This window searching strategy ensures the proximity of the forest and non-forest pixels to pixels that underwent forest change, making the estimated potential effect more representative of the actual forest change impact. To test the sensitivity of ΔCloud to window size and time period, ΔCloud was also estimated using alternative window sizes: 11 × 11 (0.55° × 0.55°), 21 × 21 (1.05° × 1.05°), 51 × 51 (2.55° × 2.55°) pixels and different periods (2002–2007, 2008–2013, and 2014–2018). The resulting ΔCloud was similar to results with the window size of 9 × 9 (0.45° × 0.45°) and among split time periods (Supplementary Figs. 2, 3). Unlike using direct comparison in cloud cover (and other biophysical variables) between forest and non-forest, an alternative method is to utilize the regression coefficients of cloud cover (dependent variable) to land cover fraction (independent variable) and estimate cloud effects assuming 100% land conversion, as adopted by ref. [58]. The alternative regression-based approach is mathematically more complicated, and its implementation involves non-trivial post-processing compared with our method while producing qualitatively similar results.

A similar window searching strategy was applied to estimate the differences between forests and non-forest in LST (ΔLST), ET (ΔET), and soil moisture (ΔSM) (Supplementary Fig. 10).

The cloud impact estimated as the cloud differences between forest and non-forest could be confounded by their differences in topography, which is known to be an important factor for cloud formation. To minimize the topographic influence, we calculated the standard deviation (s.d.) of elevation within each moving window and removed samples with s.d. >100 m from the analysis. This filtering effectively excluded comparison samples over complex terrain such as mountainous regions so that the retained samples came from relatively flat areas.

The actual effect of forest loss on cloud (ΔCloud_loss) was quantified as the cloud cover difference between forest loss (Cloud_loss) and nearby unchanged forest pixels (Cloud_forest) using the same window searching strategy as the potential effect (Eq. 2).

$$\Delta\text{Cloud}_{\text{loss}} = \text{Cloud}_{\text{loss}} - \text{Cloud}_{\text{forest}} \tag{2}$$

where $\Delta\text{Cloud}_{\text{loss}}$ is the actual impact of forest loss on cloud cover, $\text{Cloud}_{\text{loss}}$ and $\text{Cloud}_{\text{forest}}$ are the multiyear or yearly mean cloud cover averaged over forest loss and unchanged forest pixels, respectively. The actual impact (deforested vs. forests) shows good spatial resemblance to the potential effect (non-forest vs. forests, ΔCloud with the reversed sign), suggesting that the potential effect can provide a priori prediction of possible cloud change induced by forest loss (the correlation of the spatial pattern is 0.44, $p < 0.05$).

To quantify the progressive tree cover changes caused by forest loss, we calculated tree cover differences between forest loss and unchanged forest pixels following Eq. 3,

$$\Delta\text{Tree}_{\text{year}} = \left(\text{Tree2000}_{\text{loss}} - \text{Tree2000}_{\text{forest}}\right) + \sum_{2001}^{\text{year}} \left(\text{Treeloss}_{\text{loss}} - \text{Treeloss}_{\text{forest}}\right) \tag{3}$$

where $\Delta\text{Tree}_{\text{year}}$ is the tree cover difference between forest loss and unchanged forest pixels at a given year. It is the sum of the tree cover difference in the baseline year 2000 ($\text{Tree2000}_{\text{loss}} - \text{Tree2000}_{\text{forest}}$) and the accumulated yearly forest loss differences from 2001 until a given year (the sigma term of Eq. 3).

The comparison samples obtained from the window searching strategy for potential and actual impacts were aggregated to 0.5° for display and further analysis.

**Cloud effects of forests separated into different cloud types.** By using cloud top pressure and cloud optical depth from the daily Sentinel-5P NRTI data, nine cloud types were classified according to the ISCCP (International Satellite Cloud Climatology Project) cloud classification scheme[30]. The classified cloud types were 1-cirrus, 2-cirrostratus, 3-deep convection, 4-altocumulus, 5-altostratus, 6-nimbostratus, 7-cumulus, 8-stratocumulus, and 9-stratus. Cloud types 1–3, 4–6, and 7–9 corresponded to low, mid-, and high-clouds, respectively. Cloud types 3, 7, and 8 were convective clouds and the latter two were shallow convective clouds. The multiyear mean JJA total cloud fraction and fraction of each cloud type were calculated during the available time period and were aggregated to 0.05° from the original 0.01° resolution. We then applied the same moving window method to estimate the cloud effects of forests for total cloud cover as well as for different cloud types. The summed cloud effects of each cloud type equaled the total cloud cover effects. We expected convective cloud types (types 3, 7, and 8) to be influenced by forests, while other non-convective cloud types would not, so that their ΔCloud would show a more random pattern. The dominant cloud type for cloud effects of forests was determined by the cloud type whose ΔCloud had the same sign as the total cloud effect and had the largest magnitude (Supplementary Fig. 7).

We noted that there were regional differences in the cloud effects estimated from Sentinel-5P and the magnitude of the effect was also smaller than the other two datasets. For example, the Southeast US in MODIS was dominated by negative ΔCloud (64.67%) whereas in Sentinel-5P it showed more positive ΔCloud (57.09%) (Supplementary Fig. 13). The large spatial coverage of positive ΔCloud in Europe in MODIS and MSG was slightly reduced with Sentinel-5P. These regional differences might be linked to potential bias in cloud fractions of Sentinel-5P, because we found that cloud fractions of Sentinel-5P were systematically lower than that of both MODIS and MSG. However, the cloud effects of Sentinel-5P in the Amazon were consistent with MODIS (80.95%) in terms of coverage, showing a prevailing cloud inhibition (81.45%) (Supplementary Fig. 13). Cloud inhibition in Central Africa with a spatial coverage of 51.84% was slightly more in line with the widespread negative ΔCloud in MSG (67.56%) than in MODIS (36.40%).

Given these differences in the cloud effects among datasets, the results from Sentinel-5P still provided strong support that convective clouds dominated the cloud effects of forests at both global and regional scales (Supplementary Fig. 7).

**Attribution of cloud effect of forests**. Since cloud effects of forests may result from contributions of both vegetation properties and orography, we used tree cover and elevation as indicators to represent each of their effects. Elevation was selected as an indicator of the orographic lifting mechanism. We acknowledge that the reality is much more complicated than this highly simplified representation of the orographic cloud effect. However, for a global-scale analysis, elevation could still provide a first-order approximation of the orographic effect.

To decompose the potential cloud effect of forests into contributions from tree cover and elevation, we first estimated sensitivities of cloud cover to tree cover and elevation respectively, following a linear regression model defined in Eq. 4.

$$Cloud = S_{tree} \times tree + S_{ele} \times elevation + c \qquad (4)$$

where $S_{tree}$ and $S_{ele}$ were the sensitivities of cloud cover to tree cover and elevation respectively, and the intercept c was unused in this study. The sensitivity parameters were estimated for each moving window separately if it had a nonzero tree cover. The estimated slope of cloud cover to elevation ($S_{ele}$) was positive in the majority of the world (Supplementary Fig. 6d), suggesting that a higher elevation indeed promotes cloud formation. Next, we calculated tree cover differences (ΔTree) and elevation differences (ΔEle) between unchanged forests and non-forest pixels similarly to ΔCloud. Then the cloud differences induced by tree cover (ΔCloud$_{tree}$) and by elevation (ΔCloud$_{ele}$) can be obtained by multiplying their sensitivities by the corresponding differences as Eqs. 5 and 6. The sensitivity and differences parameters were averaged to 0.5° resolution before using in Eqs. 5 and 6 (Supplementary Fig. 6).

$$\Delta Cloud_{tree} = S_{tree} \times \Delta Tree \qquad (5)$$

$$\Delta Cloud_{ele} = S_{ele} \times \Delta Ele \qquad (6)$$

The reconstructed ΔCloud given by the sum of ΔCloud$_{tree}$ and ΔCloud$_{ele}$ explained about 70% of the original ΔCloud.

To attribute ΔCloud to tree cover and elevation-induced cloud changes, we compared the sign and magnitude of original ΔCloud, ΔCloud$_{tree}$, and ΔCloud$_{ele}$. If ΔCloud$_{tree}$ and ΔCloud$_{ele}$ both had the same sign as ΔCloud, the one with greater magnitude was classified as the dominant factor. If only one of ΔCloud$_{tree}$ and ΔCloud$_{ele}$ had the same sign as ΔCloud, the factor with the same sign was classified as the dominant factor. If neither ΔCloud$_{tree}$ nor ΔCloud$_{ele}$ had the same sign as ΔCloud, the dominant factor was classified as others. As a result, the potential cloud effects could be attributed to five classes: tree cover induced cloud increase (Tree+) and decrease (Tree−), orography induced cloud increase (Orography+) and decrease (Orography−), and others.

**Linking cloud effect with sensible heat flux**. Sensible heat data were obtained from three independent sources: satellite estimate[4], a Community Land Model version 5 simulation[75], and 30 paired forest and non-forest flux sites[35].

Satellite estimates provide changes in the combined sensible heat and ground heat fluxes (H+G) under different land cover conversions at 1° spatial resolution based on MODIS data (a total of 45 pairs of land conversions for "HG_IGBPdet"). The combined fluxes of H+G were estimated as the residual of surface energy components as described in ref. 4. Due to the small contribution of G to H+G, we referred to "H+G" as "H" for simplicity in the following text and the main text. To obtain sensible heat differences between forest and non-forest (ΔH) that are compatible with ΔCloud, we extracted the dominant land cover type for unchanged forest (e.g., evergreen broadleaf) and non-forest pixels (e.g., crop) within each moving window from the ESA land cover product. The dominant land cover types for forests and non-forest were upscaled to 1° resolution with the "major" method (figure not shown for 1°, but a similar one for 0.5° is shown in Supplementary Fig. 9). For each one-degree grid box with a dominant forest type (e.g., evergreen broadleaf) and non-forest type (e.g., crop), ΔH can be extracted from the corresponding sensible heat change value that matches the specific land conversion (e.g., evergreen broadleaf to crop) at the same grid box from the 45 pairs of land cover conversions defined within the "HG_IGBPdet" dataset.

CLM5 is the land component of a state-of-the-art earth system model Community Earth System Model 2 (Ref. 34). The CLM5 simulation was conducted

at the spatial resolution of 0.5° from 1997 to 2010, driven by a revised climatology GSWP3 as the atmospheric forcing (http://hydro.iis.utokyo.ac.jp/GSWP3/), with the plant phenology prescribed from satellite products, the land cover of 2000, and the separated soil columns configuration[76,77]. The years 1997 to 2001 were the spinup period and excluded from the analysis (please see detailed description in ref. [75]). In CLM, different types of vegetation within a grid cell are represented as separated tiles of different plant functional types (PFTs). We used subgrid PFT-level model outputs to calculate sensible heat differences between different land cover types within the same model grid. The subgrid tiles within a model grid cell share the same atmospheric forcing, therefore replicating the assumption of similar meteorological conditions of the space-for-time approach[12]. To match the CLM5 model resolution, the dominant land cover types for forests and non-forest of each moving window were upscaled to 0.5° using the ESA land cover data (Supplementary Fig. 9). Because CLM adopted a different land classification scheme, we created a look-up table to convert CLM land cover to the IGBP classification scheme (Supplementary Table 3). The differences in the sensible heat flux (ΔH) between a specific forest and a non-forest type can be extracted from the sensible heat values of the corresponding PFTs.

A total of 30 paired flux sites were used in this study to calculate sensible heat differences between forest and non-forest (ΔH). Twenty-eight site pairs were processed by ref. [35] using FLUXNET data and two additional Amazon site pairs were from the ORNL archive[78] (Supplementary Table 1). ΔH was calculated as the mean sensible heat flux difference between the paired forest and non-forest site during the daytime (8:00 to 16:00). ΔCloud for each site pair was extracted from the central location of the line linking two sites. Unlike ΔCloud used in the main analysis which was aggregated to 0.5°, we here used ΔCloud aggregated to 1° without the elevation s.d. criteria and the one-pixel water buffer removal to increase available ΔCloud value for each site pair. When analyzing the relationship between ΔH and ΔCloud, two flux pairs were excluded because the matched ΔCloud was missing (pair 29) and an outlier in ΔH (pair 22 with ΔH > 200 W/m$^2$).

**Scale-dependency of potential cloud effect of forest**. To investigate how the potential cloud effect varies with spatial scale, we reprocessed the MODIS cloud cover and GFC data into different spatial resolutions to emulate the scale change (using "mean" for cloud cover and "major" method for forest cover). Specifically, the 0.05° cloud and GFC data used in the main analysis were aggregated to coarser resolutions (0.1°, 0.25°, 0.5°, and 1°) and ΔCloud was re-estimated with the window searching strategy of slightly different configurations to accommodate the resolution change (Supplementary Fig. 12). The specific parameters of the window searching strategy under different resolutions are provided in Supplementary Table 2, including raw data resolution, window size, window distance, and display resolution. For a given resolution, ΔCloud was estimated with two-parameter combinations to ensure the robustness of the results.

## Data availability

All processed data that support the findings are available at Figshare[79]. The MODIS cloud cover date are available at https://lpdaac.usgs.gov/products/myd09gav006/. The MSG cloud cover data are available at https://wui.cmsaf.eu/safira/action/viewProduktSearch?menuName=PRODUKT_SUCHE/. The Sentinel-5P Near Real-Time (NRTI) data are available at https://sentinel.esa.int/web/sentinel/user-guides/sentinel-5p-tropomi. The forest cover change data are available at https://data.globalforestwatch.org/documents/14228e6347c44f5691572169e9e107ad/explore. The MODIS land cover products can be found at https://lpdaac.usgs.gov/products/mcd12q1v006/. The ESA land cover products are available at http://maps.elie.ucl.ac.be/CCI/viewer/download.php. The MODIS LST data are available at https://lpdaac.usgs.gov/products/myd11a1v006/. The MODIS ET data are available at https://lpdaac.usgs.gov/products/mod16a2v006/. The DEM data can be accessed from https://lpdaac.usgs.gov/products/astgtmv003/. The soil moisture data come from https://www.climatologylab.org/terraclimate.html. The FLUXNET data are available at https://fluxnet.org/. The AMEIFLUX data are available at https://ameriflux.lbl.gov/. The ORNL flux data can be found at https://daac.ornl.gov/cgi-bin/dsviewer.pl?ds_id=1174. Alternatively, the MODIS, Sentinel, Forest cover change, and Soil moisture data are readily accessible on Google Earth Engine (https://earthengine.google.com/).

## Code availability

All codes of this study are available at Figshare[79].

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

## Acknowledgements

This study is supported by the National Key Research and Development Program of China (No. 2017YFA0604701) and the Fundamental Research Funds for the Central Universities. RX is supported by the Postdoctoral International Exchange Program. We would like to thank the high-performance computing support from the Center for Geodata and Analysis, Faculty of Geographical Science, Beijing Normal University [https://gda.bnu.edu.cn/].

## Author contributions

Y.L. conceived and designed the study; R.X. and Y.L. performed the data analysis; Y.L., R.X., A.J.T., and L.Z. analyzed the results, with help from D.V.S., L.G.-C., R.M., L.C., Y.Z., and B.F. in the interpretation of the results; Y.L. and R.X. wrote the manuscript with contributions from all authors. RM conducted the CLM5 simulation; LC and HL provided and processed the flux tower data.

## Competing interests

The authors declare no competing interests.
