## [Peer Review File · Nature Communications]

Contrasting impacts of forests on cloud cover based on satellite observationsREVIEWER COMMENTS

Reviewer #1 (Remarks to the Author):

Comments to the Authors:

In their manuscript "Contrasting impacts of forest on cloud cover based on satellite observations", the authors present an analysis of satellite cloud observations and relate changes in clouds to (a) historical changes in vegetation over the satellite era and (b) differences in land-cover type (forest vs non-forest) in close proximity. The authors find that in some places, forests favor cloud formation, while in other places, forests seem to hinder cloud formation. There is a growing literature exploring how terrestrial vegetation may be impacting cloud cover, and this is an exciting topic. While models show cloud responses to vegetation change, one of the major challenges in this area of research is to detect a signal in real-world data, which the authors attempt to do here.

While I commend the authors on tackling this tricky problem, I have reservations about the study and the conclusions the authors draw which would need to be addressed prior to the manuscript being suitable for publication. I have outlined these concerns below.

Major (scientific/methodological):

- Context for the magnitude of cloud changes detected in the satellite observational record needs to be provided. The authors talk about "delta Cloud", the "potential cloud effect", and "delta cloud of +0.0133" etc., but don't tell us what this means! Change in cloud fraction within a gridcell? Within a moving window? Fraction of days where clouds were detected vs clear days? W/m² radiative effect? Something else?

- The authors don't discuss how the advection by low-level winds of air (and moisture) from forested gridcells to non-forested gridcells (and vice-versa) could impact the correlations they're looking at here. Are there big cloud changes in places with large-ish surface winds/advection? How would the resolution considered alter this? Doing something to demonstrate in which regions advection of clouds or cloud-forming air from one land type to another are an important player in the story, or showing that this doesn't matter, seems important.

- Generally speaking, this paper has a *lot* of supplemental figures and leans heavily on them to tell the story. This makes it hard to follow the paper without keeping the supplement open at the same time, which means those figures really ought to be in the paper. This probably makes the paper too long for this journal's format, so I understand why the authors may have moved them to the supplement, but the purpose of the supplement isn't just as overflow for what didn't fit in the paper. Can the authors try and tell a story just with what they can show in the paper?

- The agreement between the CLM and satellite-derived "potential forest effect on sensible heat flux" isn't very good – what should the reader take away from this? (figure 3) What does the comparison with CLM add given the poor agreement?

Specific comments:

- The manuscript would benefit from a read-through with attention to grammar, e.g.

- Line 32: "forests can have opposite effects on summer cloud cover" – based on location/climate? Otherwise this is a confusing sentence – could mean "one day the forest effect was X, the next day it was Y"

- Line 53: "results in line with the common perception that vegetation enhances clouds and precipitation" – I don't think this is a common perception? I spend most of my time thinking about plants and clouds, and I don't expect more plants to necessarily = more clouds... I generally do in

the tropics if we're comparing a rock to a patch of vegetation, but the tradeoff between more sensible heat over vegetation with low ET (thus deepening the boundary layer) vs more latent heat over vegetation with high ET (thus increasing boundary layer humidity) could both help (or hinder) a parcel of air to reach the LCL. Outside of the tropics, the response is even more complicated, and is dependent on soil moisture, atmospheric stability, sensible vs latent heat partitioning, how close the moist the atmosphere blowing into the region is, etc.

- Line 58: "hardly comparable with observations" - While I agree that GCM simulations with large imposed changes in vegetation result in climate changes due both to the local vegetation change and the remote vegetation change, I don't agree that that isn't a problem in observations. Observations also can't separate if the atmospheric change observed over some delta land cover is driven by that local delta land cover, by some remote delta land cover (e.g. the Winckler et al 2019 paper the authors site here had large "remote" effects come from neighboring gridcells), or by some remote completely different forcing.

- I don't understand figure 1b (or d). Do the authors take the positive gridcells and zonally average them, then take the negative gridcells and zonally average them? What does this tell us? What is the actual mean of all grid cells? What is the point of separating them (I assume there is a point, or they wouldn't have done it, it just wasn't very clear anywhere what the reason was - so perhaps the authors could add some text here to explain the motivation of considering only the negative gridcells, and only the positive gridcells (separately) at each latitude).

- Line 83: please clarify, is one cell 0.45x0.45, or is the 9x9 window 0.45x0.45 (I assume the latter given the MODIS resolution, but please clarify)

- Line 85: If the cloud change is very remotely driven - but this won't account for, e.g. advection between neighboring patches. How do the authors account for that, or determine that they don't need to account for that? (related to the wind/advection comment above)

- Lines 88-89: I'm not saying don't look at JJA, but worth specifying that JJA has been shown to have maximal cloud responses over forests in *Europe* - one region of the NH Mid-Latitudes. One of the novel aspects of your study is that you're applying your analysis globally, and JJA is not "summer" / summer-like over the whole globe - particularly the tropics where seasonality is quite different, or the SH mid-latitudes where you're effectively looking in winter. There is more land in the NH, so you probably can keep your analysis mostly focused on JJA, but a discussion of "actually we'd expect DJF to probably have a stronger signal in the SH mid-latitudes, and indeed we find that / or we don't find that" would be appropriate. Additionally, wouldn't you expect the largest impact of forests on clouds (or at least on column moisture) over the much of the tropics (esp. the Amazon) to be during the dry / dry-wet transitional season, which is not necessarily JJA (location dependent)? (See Spracklen et al 2012, doi.org/10.1038/nature11390)

- Line 93: ie forests have less cloud cover than their non-forest neighboring gridcells? This seems opposite of what I would expect re: the next sentence, when the authors say "due to strong turbulent flux contrast between forest and non-forest in low lats" - can you show the contrast is stronger in low lats than high lats? Wouldn't stronger turbulence = cloud *enhancement* over forests? Some reworking of the text here might clear up my confusion.

- Line 92: "+0.0133" - what is this number? A change in ... cloud frequency? Fractional cloud cover? It is unclear from lines 81-82 what "Cloud" numerically means in this analysis.

- Line 96-97: the fact that there are large enhancements and inhibitions of clouds over forests in the tropics means a single explanation (like is made in the previous sentence" is unlikely to explain the response. It isn't like the cloud response is all one sign and the tropics and all another sign in the high lats, or I'm misreading figure 1b. I don't find figure 1 very supportive of the claims made here.

- Line 102-103: "inhibition stronger in central Africa, weaker in the Amazon regions..." Why? And what about the strong mix of positive and negative signals indicated in figure 1?

- Line 116-117: "cloud inhibition is stronger during the dry season in the Amazon, but amplified during the wet season in Central Africa" - why? without presenting some potential explanations for why two tropical forests at the same latitude would produce opposite cloud responses, these correlations don't provide particularly compelling evidence for the cloud responses being driven by the forests. I'm not saying they're not - they very well could be, and the seasonality and climatology of the Amazon vs. central African rainforests have some distinct differences. So, it would strengthen the paper to discuss how those differences could physically be expected to produce different responses of clouds to forest change, and then discuss if your results do/do not support those physical mechanisms, would be much more compelling.

- Figure 2: MSG and MODIS don't appear to agree on the attribution here. MSG is largely blue across central Africa, MODIS is largely red/rainbow. Please explain this disagreement.

- Lines 126-127: Would it not be a more accurate representation of ref. 27 to say that "human-causes forest loss tends to be stronger at lower elevations and in regions of non-complex terrain"? Plenty of forests are located in regions of simple terrain and low elevation, and plenty more would be there if people didn't cut them down. The way this is written right now is just misleading in that it makes it seem like forests preferentially grown on steep mountain slopes, while actually they would happily grow many other places, they're just easier to cut down in many of those other places.

- Line 129 & figure S7: Some clarification here is needed, I think - to explicitly state that orography (when moist air is pushed up the slope, anyhow) promotes cloud formation, and the fact that many forests are left in regions of complex topography might then result in the forests being under clouds simply because they are on a slope, and not because they are a forest... I think that is what the authors are trying to say here, it just wasn't super clear. Similarly, I was confused looking at figure S7 what it was trying to convey. I think the idea was supposed to be that the forest on the hill has clouds because it is on a hill, not because it is a forest. In that case, maybe replacing the forest in (b) with grass would be more helpful, and even adding 2 more panels where the hill is flattened and with a schematic of how the forest would impact clouds *in the absence* of a hill might be helpful.

- Figure 3b: What are the circles for? (add to caption)

- Figure 3c: What are the lines? The paired sites? If so, they're pretty far away. If not, what are they? Why do only some sites have lines? The caption says the lines indicate flux tower clusters, but ... is a cluster the same as a paired site, or is it a place where there are lots of paired sites?

- Figure 3d: would this be considered a statistically significant slope? I'm more familiar with the Pearson correlation test (I'm not at all saying it is better/worse in this case), but doing a quick search of the Spearman's correlation test doesn't tell me how its p-value is calculated. Is it the p-value from a student's t-test? If so, please add that to the caption to explain. If not, what is it?

- Figure 3: What is "potential effect of forest on sensible heat"? The paper would benefit from more clearly laying out the definitions for the metrics the authors use.

- Line 141: "One pair in the Amazon is not shown on the map" - why?

- Line 150-151: "forest has ... lower land surface temperature, increased evaporation and soil moisture than nonforest vegetation". Not always. The authors could hedge here and say "generally...". But Some agricultural areas in particular have higher fluxes than forests, and seasonally forests can have really low ET (e.g. in the spring in the high latitudes when the deeper soils are frozen) compared to grasses/shrubs with shallow root systems. However, I don't think the authors need to get into a detailed deep-dive on nuanced forest vs non-forest flux responses under specific conditions, seasons, etc. - rather, they could simply soften the statement they make a bit so as not to distract readers like me who will see and immediately ask "but what about this special case?"

- Lines 155-158: Figure 3c is just from the paired flux towers, isn't it? Is the slope robust if you use CLM and satellite observations, too? It seems like 3c should be showing the regression for all methods used, because 3b is just showing the change in H, not how the change in H correlates with (or doesn't) the change in clouds. It is unclear what b means by "satellite" (MODIS? MODIS and MSG?), but if it is MODIS, the reader could try and mentally regress delta H from b to delta cloud in a, but delta cloud is never shown for CLM.
- Lines 158-160: I'm pretty confused what is being shown in 3b. Did you take forested gridcells and nearby non forested gridcells and difference the H between those?
- Lines 167-171: I'm confused at what the authors are getting at here; this seems sort of like saying what the Bowen ratio is twice. It isn't because the Bowen ratio is small that LH is comparatively large and SH is comparatively small. That is the definition of the Bowen ratio. So, LH is big and SH is small therefore the Bowen ratio is small seems redundant... I don't understand what they're getting at.
- Line 170: This is the closest the authors get so far to saying anything about how vegetation influences boundary layer development and the lifting of moist parcels of air to the LCL. Which seems like a critical part of the physical argument. The authors should consider devoting more space to this topic.
- Lines 179-195: This is a clear and nice summary, though is more of a lit review than a "results from this study". Maybe the authors could weave in what their results add to the state-of-the-science here. In addition, including a reference/discussion of how the authors' results relate to the literature exploring the response of clouds and precipitation to soil moisture regimes would be beneficial here, as it is related to the argument as they show in figure 4 (except being driven by sensible/latent heat partitioning as a function of soil moisture).

Possibly useful references from that section of the literature:

Ek, M. B., & Holtslag, A. A. M. (2004). Influence of soil moisture on boundary layer cloud development. *Journal of Hydrometeorology*, 5(1), 86–99. [https://doi.org/10.1175/1525-7541\(2004\)005<0086:IOSMOB>2.0.CO;2](https://doi.org/10.1175/1525-7541(2004)005<0086:IOSMOB>2.0.CO;2)

Welty, J., Stillman, S., Zeng, X., & Santanello, J. (2020). Increased Likelihood of Appreciable Afternoon Rainfall Over Wetter or Drier Soils Dependent Upon Atmospheric Dynamic Influence. *Geophysical Research Letters*, 47(11), 1–9. <https://doi.org/10.1029/2020GL087779>

Guillod, B. P., Orlowsky, B., Miralles, D. G., Teuling, A. J., & Seneviratne, S. I. (2015). Reconciling spatial and temporal soil moisture effects on afternoon rainfall. *Nature Communications*, 6(March), 1–6. <https://doi.org/10.1038/ncomms7443>

Koster, R., Dirmeyer, P. A., Guo, Z., Bonan, G., Chan, E., Cox, P., ... Vasic, R. (2004). Regions of Strong Coupling Between Soil Moisture and Precipitation. *Science*, 305, 1138–1140.

Dirmeyer, P. A. (2011). The terrestrial segment of soil moisture-climate coupling. *Geophysical Research Letters*, 38(16), 1–5. <https://doi.org/10.1029/2011GL048268>

Koster, R. D., Guo, Z., Dirmeyer, P. A., Bonan, G., Chan, E., Cox, P., ... Yamada, T. (2006). GLACE: The Global Land-Atmosphere Coupling Experiment. Part I: Overview. *Journal of Hydrometeorology*, 7(4), 590–610. <https://doi.org/10.1175/JHM510.1>

- Figure 5: These trends, except maybe over the Maritime Continent, are very small (and for the Maritime Continent, showing it is forest loss and not ocean variability that is driving the signal would be necessary). Could the authors give some context to the magnitude of the changes and slopes they're showing here?
- Lines 229-231 (and figure 5): some indication of statistical significance is necessary, because these relationships are extremely small.

- Line 232: I don't think figure 5f shows this. 5f just shows the study period, or I'm missing something. Please clarify.

- Can the authors comment on the biases in their ET dataset?

- Figure S4: the color maps in both (a) and (b) are extremely hard to me to tell apart. I look at (a) and just see "orange-ish everywhere", rather than a clear "altostratus" vs "stratocumulus" vs "others". Similarly in (b) the bars are just a bit hard on the eyes. Even adding a black outline to each bar here would probably help a lot.

- Figure S5: what *area* of each region (as a weight - e.g. 10%, or 80% etc) show positive vs negative effects? Including such weighting in this figure, and in the figures in the main text, on the line plots of both positive and negative effects would help the reader know if there are equal and opposite magnitudes of positive and negative response, or if the areas of positive response are much smaller than the areas of negative response (or vice versa)

- Figures S2, S3, S8, and generally throughout the paper – please make it clear what “delta cloud” is (e.g. change in cloud fraction)

Minor comments:

- Line 35-36: this usage of brackets is confusing. Could write the sentence only slightly longer but much more clearly as:

"driven by sensible heating where cloud enhancement is more likely to occur when sensible heat over the forest is larger than over nearby non-forested regions, and cloud inhibition occurs when sensible heat over the forest is smaller."

See the below EOS article from 2010 on the subject – I'm not alone in finding this confusing!

<https://eos.org/opinions/parentheses-are-not-for-references-and-clarification-saving-space>

- Line 37: “opposite cloud cover changes” – opposite of what?

- Line 43: “processese” – do the authors mean processes?

- Line 62 – “inhibited clouds over SOME forest”

- Figure 1 - I would suggest the authors reverse their color bar here, such that blue = more clouds and red = less clouds. Intuitively, red suggests "dry" and blue suggests "wet", and since clouds are a moisture related field this could be helpful.

- Figure 1e – again, the authors might want to consider a different set of 4 colors. It is extremely hard for my eyes to tell the difference between the cyan and green colors used here.

- Line 77: a brief introduction to MSG and MODIS is necessary since the methods appear at the end of papers in these formats. Just "satellites that detect clouds" is good enough.

- Line 82: why is (spatial) in brackets?

- Line 99: “splited” -> “split”

- Line 101: necessary to say something about MSG getting hourly resolution but non-global coverage (could say this when you briefly introduce MOIDS and MSG)

- Line 143: So the cloud fraction for the tower sites was taken from MODIS? (Just a clarification question here)

- Line 157 (ref 31) - is this the citation the authors meant to use? I thought this study looked at accounting for biomass heat storage on diurnal temperatures... could the authors elaborate on why that is the appropriate CLM simulation to explore here, and how exactly they're calculating delta H (using a similar forest vs nonforest site nearby like in the satellite observations?) Why not use a CLM simulation with deforestation?

- Line 164-165: bracket usage

- Line 167: “levelsl” -> “levels” (typo)

- Line 192: brackets

Reviewer #2 (Remarks to the Author):

The authors present an investigation of cloud cover differences between forests and nearby unforested areas and find regionally different behavior of cloud enhancement and inhibition.

In general, the topic is important and of interest to the community, given the fact that changes in forest cover are likely accompanied by changes in cloud cover and potentially precipitation. In the light of global change a better understanding of the direction of these processes and the underlying mechanisms would be appreciated.

I have several major comments that I think are important to address before publication.

1. Proposed mechanisms for cloud cover change and confidence.

a) The authors present a mechanism based on differences in sensible heat flux. At the same time, figure 3b does not show very strong relationship as indicated by the p-value and the fact that most sites seem to show a Δ_{cloud} in the range of 0-0.02 irregardless of Δ_H . Also, Figure S12 shows that MODIS and Sentinel do not agree on the sign of the effect in 2 of the 4 regions. It appears that there is a clear signal in the Amazon and in EurAsia, but not in the US and Central-Africa. I feel that caution should be taken in explaining cloud cover differences based on a mechanism if we cannot establish the overall sign of the effect for a given region depending on the data used.

b) The mechanism proposed by the authors is established in Figure 4. Based on figure 4a, the authors indicate that forests with enhanced cloud cover have both higher H and LE compared to non-forested regions, which then leads to higher ABL and more LCL crossings. I am a bit confused by this since, sensible and latent heat fluxes together tend to balance the net radiation.

Despite the difference in albedo, I am not sure that I would expect sensible heat fluxes in non-forested environments to be lower than forest H (especially in tropical regions) and forest ABLs to be higher. For example Fisch et al (2004) cited by the authors does not show ABL height differences during the wet season and higher ABL and H over pasture compared to forest during the dry season, which contradicts the authors stated mechanism for Amazonia.

Given the fact that cloud cover itself affects the energy supply, I feel that a deeper dive into the data is needed to 'prove this relationship'. For boreal regions, it may also not be true, since I would expect wetlands to have a similar (or even lower albedo) compared to forests. The authors should use the data to also look into ET/LE which is not presented here.

I feel that the mechanisms shown in figure 4b is fairly well established in the literature.

c) The differing behavior of EurAsia may be due to differences in the land-cover pairs. Based on Figure S9 a large portion of the nonforested land cover in the boreal forests is wetlands, which have a very different surface energy balance compared to grasslands and croplands, which are found in the other regions. It may thus make sense to formulate 2 mechanisms rather than trying to combine everything into a single theory. It may make sense to limit the scope of the paper to tropical regions (especially since there seems to be a clear signal in the Amazon irregardless of the cloud cover sensor).

d) I am wondering to what extent cloud cover as derived by Modis is a good category for analysis, since it is a simple yes/no mask with no additional information related to type of cloud, cloud water, fractional cover, etc. The Sentinel data seems to provide a much richer set of information, but also complicates the analysis tremendously.

2) Organization of the manuscript

The attribution section feels out of place. I would consider this data preparation and would move this to the methods section and then only use the data without potential orthographic effects.

3) Presentation of the manuscript

a) The manuscript relies heavily on maps with many small pixels, that show substantial small scale variation (e.g. Figure 1a). It is very difficult to extract quantitative and even qualitative information from these maps. I would encourage the authors to think, how they can synthesize this information into better formats along the lines of Figure 1b, 3b, Figure S12 etc.

b) The authors should ensure that abbreviations are introduced in the text (and not only in the methods).

We deeply appreciate the detailed and constructive comments provided by the two anonymous reviewers. Following their suggestions and comments, we have extensively revised the manuscript and provided a point-to-point response to each comment. We believe that the manuscript has been much improved. The original comments are in **bold** font, our response is in regular font, and the changes in the text are in blue.

In their manuscript “Contrasting impacts of forest on cloud cover based on satellite observations”, the authors present an analysis of satellite cloud observations and relate changes in clouds to (a) historical changes in vegetation over the satellite era and (b) differences in land-cover type (forest vs non-forest) in close proximity. The authors find that in some places, forests favor cloud formation, while in other places, forests seem to hinder cloud formation. There is a growing literature exploring how terrestrial vegetation may be impacting cloud cover, and this is an exciting topic. While models show cloud responses to vegetation change, one of the major challenges in this area of research is to detect a signal in real-world data, which the authors attempt to do here.

While I commend the authors on tackling this tricky problem, I have reservations about the study and the conclusions the authors draw which would need to be addressed prior to the manuscript being suitable for publication. I have outlined these concerns below.

We thank this reviewer for commending the significance of our study, and we have made significant efforts to address these concerns in the revision.

Major (scientific/methodological):

- Context for the magnitude of cloud changes detected in the satellite observational record needs to be provided. The authors talk about “delta Cloud”, the “potential cloud effect”, and “delta cloud of +0.0133” etc., but don’t tell us what this means! Change in cloud fraction within a gridcell? Within a moving window? Fraction of days where clouds were detected vs clear days? W/m² radiative effect? Something else?

The cloud effect in our study is measured by cloud cover fraction, which is calculated from daily cloud mask data at 1km for MODIS and 0.05 degree for MSG, indicating the occurrence frequency of clouds over a period of time with a valid range of 0-1.

In the revision, we added an explanation for the measurement of cloud effect at the beginning of the results section, and provided a supplementary figure for cloud cover fraction data. Please see the revised text and figure below.

Here, the cloud effect is measured by cloud cover fraction derived from Moderate Resolution Imaging Spectroradiometer (MODIS) and Meteosat Second Generation (MSG) satellite data, which represent the occurrence frequency of clouds over a period of time with a valid range of 0-1 (Fig. R1).

Fig. R1 (Figure S1). (a) The spatial distribution of multi-year mean MODIS JJA cloud cover fractions at 0.05° from 2002 to 2018. (inset: MSG JJA cloud cover fractions from 2004 to 2013). (b) The standard deviation of the time series of MODIS JJA cloud cover fraction during 2002-2018 and (c) its latitudinal pattern.

With this measure, the potential cloud effect is defined as the multiyear mean cloud cover differences between unchanged forest and nearby nonforest pixels ($\Delta\text{Cloud} = \text{Cloud}_{\text{forest}} - \text{Cloud}_{\text{nonforest}}$). Figure 1 shows ΔCloud estimated across the globe using the moving window after aggregating to 0.5° for display. The reported ΔCloud of $+0.0133$ in the initial manuscript means the globally averaged magnitude of cloud enhancement effect. In other words, for grid samples with positive ΔCloud , the cloud cover fractions in forests are on average 0.0133 larger than non-forest. We revised the description of these results to avoid confusion and the revised texts are shown below:

Most temperate and boreal forests in Eurasia and North America have higher cloud fractions than non-forest, indicating a cloud enhancement effect (positive ΔCloud) accounting for 63.21% of all grid samples with a global mean magnitude of $+0.0133$. In contrast, forests in South Amazon, Central Africa, and Southeast US have lower cloud fractions than nearby non-forest, signifying a cloud inhibition effect (negative ΔCloud) over the forest with a global mean magnitude of -0.0115 .

- The authors don't discuss how the advection by low-level winds of air (and moisture) from forested gridcells to non-forested gridcells (and vice-versa) could impact the correlations they're looking at here. Are there big cloud changes in places with large-ish surface winds/advection? How would the resolution considered alter this? Doing something to demonstrate in which regions advection of clouds or cloud-forming air from one land type

to another are an important player in the story, or showing that this doesn't matter, seems important.

Thanks for this suggestion. It is true that low-level advection can affect the cloud effect of the forest identified in our study. The advection driven by low-level winds could affect the strength and position of mesoscale circulation and therefore the cloud effect. This effect can be manifested by analyzing the cloud effect under different synoptic conditions, as demonstrated in Teuling et al (2017). Their results showed that convergence over the forest favours cloud development under calm conditions would change to weakened convergence at the upwind edge (due to advection of cool air into the forest) and enhanced convergence at the downwind edge under moderate winds conditions. Such analysis can be performed by studies focusing on a specific region but it is difficult for global-scale study because of different geographical characteristics. Our study focused more on global patterns at climatological scales to minimize the effect of these synoptic variations.

(1) In the revision, we added discussion on how synoptic winds affect the cloud effect of forests:

The strength and position of mesoscale circulation, as well as the resultant cloud effect, are influenced by synoptic conditions as well. Mesoscale circulations get intensified under weak synoptic conditions, because large wind flow eliminates the thermal gradient generated by land heterogeneity¹⁶. For example, the cloud inhibition effect in Amazon forests becomes stronger during the dry season when synoptic winds are weaker and the LST gradient is larger^{5,15,18}. Under moderate winds conditions, due to low-level advection, the convergence over forest could be shifted along the wind direction, causing enhanced convection downwind and suppressed convection upwind of the forest^{22,44}. The variations in cloud effects due to varying synoptic conditions are most prominent at fine temporal scales for a specific region, which may not necessarily be manifested in our global pattern derived at climatological scales.

(2) We modified the mechanistic figure to show this advection impact which pushes the convergence to downwind location.

Fig. R2 (Figure 4). Mechanisms of contrasting cloud effects of forests. (a) Clouds enhanced over forests through increased convection due to increased moisture supply and turbulence. (b) Clouds inhibited over forests through suppressed convection due to divergence of mesoscale circulations. ABL: atmospheric boundary layer. LCL: lifting condensation level. LE: latent heat. H: sensible heat.

(3) We also analyzed the global surface wind speeds but we did not observe a clear linkage to the cloud effect (Fig. R3).

Fig. R3. Global distribution of surface wind speed for JJA during 2002-2018 (Data source: TerraClimate).

(4) We feel that wind advection from one land to another land type could be an important factor but it is difficult to analyze with existing datasets, especially at small scales. For example, the most recent EAR-5 reanalysis data have a spatial resolution of 0.25 degree. Such resolution is still too coarse to capture wind advection between forest to nonforest caused by the fine-scale land heterogeneity in our analysis. Other studies looking at the relationship between soil moisture anomaly and rainfall (Taylor et al., 2012) did not find a prominent role of advection, at least not for local and daytime convection.

- Generally speaking, this paper has a *lot* of supplemental figures and leans heavily on them to tell the story. This makes it hard to follow the paper without keeping the supplement open at the same time, which means those figures really ought to be in the paper. This probably makes the paper too long for this journal's format, so I understand why the authors may have moved them to the supplement, but the purpose of the supplement isn't just as overflow for what didn't fit in the paper. Can the authors try and tell a story just with what they can show in the paper?

Thanks for the suggestion. Due to the space constraints and the journal style, we need to put many figures in the supplementary information. These supplemental figures provide complementary information to complete and strengthen the story. In the revision, we made several changes to improve the presentation of the manuscript.

- We updated the figure for the seasonal variations of the cloud effects and moved it from SI to Figure 2 in the revised main text.
- The original Figure 2 and the corresponding text about cloud effect attribution were merged to the section "Potential effects of forests on cloud cover".

With these changes, the content of the paper is reorganized as follow:

- 1) The spatial pattern of the cloud effects of forests ;
- 2) The seasonal variations of the cloud effects of forests;
- 3) The relationship between sensible heat differences and cloud effects;
- 4) The mechanisms of the enhanced and inhibited cloud cover over the forest;

5) The forest loss impacts on cloud cover in hotspots regions.

Please check the revised manuscript for more detailed changes.

- The agreement between the CLM and satellite-derived “potential forest effect on sensible heat flux” isn’t very good – what should the reader take away from this? (figure 3) What does the comparison with CLM add given the poor agreement?

Since the accurate estimation of sensible heat is still a difficult and challenging task, we used three independent datasets to estimate sensible heat differences (ΔH) between forest and nonforest (including satellite estimates, CLM, and flux towers), and each of these datasets has its own uncertainties and limitations. For satellite estimates, sensible heat is estimated as the residual energy flux ($H + G = \text{net radiation} - LE$), which includes the ground heat flux. This residual flux is affected by measurement uncertainties of all other land surface energy flux components. For the CLM estimates, land surface models are known to have biases in simulating H and ET . However, conservation of energy is warranted in CLM. Further, energy fluxes at the land surface can be retrieved under cloud-free conditions only by satellites, while the sensible heat flux difference simulated by CLM represents all conditions. Large uncertainties can be expected for both the satellite and CLM estimates of H , as indicated by their differences in spatial pattern shown in Fig. 3. Therefore, we did not expect strong quantitative relationships between ΔH from satellite/CLM and $\Delta Cloud$. Instead, their relationships with $\Delta Cloud$ are presented as the broadly spatial correspondence between cloud inhibition and negative ΔH or cloud enhancement and positive ΔH . The broad agreements between ΔH and $\Delta Cloud$ are highlighted by the three circles marked on the map of Fig. 3.

Although flux tower measurements have their own limitations (e.g., few sites, distance between sites), they are considered to be the most accurate method for estimating sensible heat among the three datasets, and are often used as the "ground truth" to validate satellite and model estimates. Therefore, we only present the quantitative relationship (scatterplot) between ΔH from paired flux towers and the corresponding $\Delta Cloud$ from MODIS in Figure 3c.

Here, in the response letter we provide scatterplots for ΔH from satellite/CLM against $\Delta Cloud$ (Fig. R4). Results support the positive relationships between these two with spearman correlation coefficients of 0.14 ($p < 0.01$) and 0.11 ($p < 0.01$) for satellite- and CLM-derived ΔH , respectively. Although the relationships are not as strong as those for flux towers (possibly due to large uncertainties in the H estimates), they confirm the positive linkage between ΔH and $\Delta Cloud$ shown in Fig. 3b.

Fig. R4. The regression between ΔCloud and ΔH from (a) satellite and (b) CLM. The correlation coefficient refers to the spearman correlation coefficient.

To sum, it is the relationships collectively confirmed by three independent datasets in either qualitative or quantitative ways that support our findings on the linkage between ΔH and ΔCloud .

We also mentioned the relationship between ΔH from satellite/CLM and ΔCloud in the revised paragraph:

Such a spatial co-occurrence is further confirmed by the positive relationship between observed ΔH from paired flux sites and MODIS ΔCloud ($\rho=0.34$, Fig. 3d), and those between ΔH derived from satellite/CLM and ΔCloud (weaker but statistically significant; not shown), suggesting that cloud enhancement is more likely to occur when sensible heat over the forest is larger than nearby non-forest, and cloud inhibition occurs when sensible heat over the forest is smaller.

Specific comments:

- The manuscript would benefit from a read-through with attention to grammar, e.g.

Thank you! We have carefully revised the manuscript in the revision.

- Line 32: “forests can have opposite effects on summer cloud cover” – based on location/climate? Otherwise this is a confusing sentence – could mean “one day the forest effect was X, the next day it was Y”

Sorry for the confusion. Here we mean the opposite cloud effect in different regions. For example, forests decrease cloud cover in Amazon while increasing cloud cover in Europe. This sentence has been revised to clarify this point:

Here, we use satellite data to show that forests in different regions have opposite effects on summer cloud cover.

- Line 53: “results in line with the common perception that vegetation enhances clouds and precipitation” – I don’t think this is a common perception? I spend most of my time thinking about plants and clouds, and I don’t expect more plants to necessarily = more clouds... I generally do in the tropics if we’re comparing a rock to a patch of vegetation, but the tradeoff

between more sensible heat over vegetation with low ET (thus deepening the boundary layer) vs more latent heat over vegetation with high ET (thus increasing boundary layer humidity) could both help (or hinder) a parcel of air to reach the LCL. Outside of the tropics, the response is even more complicated, and is dependent on soil moisture, atmospheric stability, sensible vs latent heat partitioning, how close the moist the atmosphere blowing into the region is, etc.

Sorry for the confusion. It is true that the relationship between vegetation and cloud/precipitation is complex at regional and local scales, as exemplified by the reviewer. Here we emphasize their relationships at continental and global scales, that is, vegetation accelerates the water cycle. For example, by comparing a simulated desert and green-planet world, Kleidon (2000) showed that the presence of vegetation led to a doubled precipitation and +16% cloudiness (especially in the tropics) and these changes are attributable to the higher moisture content of the atmosphere due to the increased recycling. The enhancing effect of vegetation on precipitation/cloud is found at large scales but not necessarily at small scales. Therefore, we revised this sentence as:

Although these results generally support that vegetation enhances clouds and precipitation at large-scales^{10,11}, especially in the tropics, these continental- or global-scale land clearing experiments implemented in models with a relatively coarse resolution are not consistent with the ongoing small-scale land activities in the real world.

- Line 58: “hardly comparable with observations” - While I agree that GCM simulations with large imposed changes in vegetation result in climate changes due both to the local vegetation change and the remote vegetation change, I don't agree that that isn't a problem in observations. Observations also can't separate if the atmospheric change observed over some delta land cover is driven by that local delta land cover, by some remote delta land cover (e.g. the Winckler et al 2019 paper the authors site here had large "remote" effects come from neighboring gridcells), or by some remote completely different forcing.

For observational studies, a critical step is to isolate the local and remotely non-local signals. This is typically done by comparing a site with land cover change (local + nonlocal signals) and another neighboring reference site that contains only the nonlocal signal. The differences between the site with land cover change and the reference site thus isolate the local effect with the assumption that the same non-local effect is shared by both sites (e.g., the remote impact resulting from distant land cover change, or atmospheric change). Therefore, the impact obtained from observational studies using this approach is often considered to be the local impact. As shown by Figure 1 of Winckler et al 2019, after isolating the nonlocal impacts, the resulting local effect in climate models is very similar to the local signals from observation. We hope this explanation helps clarify this.

We also acknowledged this nonlocal effect at the end of discussion as:

Moreover, the changing forest cover owing to either deforestation, increased tree vulnerability under future warming^{64,65}, or afforestation⁶² will not only affect local climate and hydrology, but also cause remote impacts on distant regions through moisture recycling and advection⁶⁶ and have other ecological and socio-economic implications^{55,67}.

- I don't understand figure 1b (or d). Do the authors take the positive gridcells and zonally average them, then take the negative gridcells and zonally average them? What does this tell us? What is the actual mean of all grid cells? What is the point of separating them (I assume there is a point, or they wouldn't have done it, it just wasn't very clear anywhere what the reason was – so perhaps the authors could add some text here to explain the

motivation of considering only the negative gridcells, and only the positive gridcells (separately) at each latitude).

The interpretation is correct. Fig. 1b,d showed the zonal averages of ΔCloud with positive and negative signs separately. This separation is necessary because our results indicated the contrasting cloud effects of forests, that is, cloud enhancement (positive ΔCloud) and inhibition (negative ΔCloud). Here we want to emphasize how their magnitudes change with latitude. If not separating these two, the contrasting cloud effects would cancel each other in the zonal average due to their opposing signs, and the resulting plot would mix their frequency and magnitude, as shown in the figure below (Fig. R5).

Fig. R5. The latitudinal pattern of ΔCloud based on MODIS without separating positive and negative values.

This paragraph has been revised to clarify the motivation of separating the positive and negative ΔCloud :

Most temperate and boreal forests in Eurasia and North America have higher cloud fractions than non-forest, indicating a cloud enhancement effect (positive ΔCloud) accounting for 63.21% of all grid samples with a global mean magnitude of +0.0133. In contrast, forests in South Amazon, Central Africa, and Southeast US have lower cloud fractions than nearby non-forest, signifying a cloud inhibition effect (negative ΔCloud) over the forest with a global mean magnitude of -0.0115. The strength of these contrasting cloud effects (i.e., cloud enhancement and inhibition) follows a latitudinal dependency with the largest magnitude in the tropical regions and diminished toward higher latitudes (Fig. 1b). This is likely due to preferential conditions for convection development at low latitudes, as indicated by their high convective available potential energy which decreases at higher latitudes²⁷.

- Line 83: please clarify, is one cell 0.45x0.45, or is the 9x9 window 0.45x0.45 (I assume the latter given the MODIS resolution, but please clarify)

The sentence has been revised to:

Δ Cloud is estimated globally through a moving window sized at $0.45 \times 0.45^\circ$ (9×9 cells) near locations that underwent forest cover change during the study period (see methods).

- Line 85: If the cloud change is very remotely driven - but this won't account for, e.g. advection between neighboring patches. How do the authors account for that, or determine that they don't need to account for that? (related to the wind/advection comment above)

If the advection is driven by synoptic winds, it would affect the strength and position of mesoscale circulation and therefore the cloud effect, as discussed in our response to the previous comment. If the low-level advection between neighboring patches is driven by their thermal differences such as those between forest and nonforest, this advection becomes part of the mesoscale circulation (the low-level branch), and contributes to the cloud difference between forest and nonforest. Because we used the multiyear mean cloud cover fraction to quantify the cloud effects, any persistent cloud effects from advection (either locally or remotely driven) will be captured while any random effects will be smoothed out.

- Lines 88-89: I'm not saying don't look at JJA, but worth specifying that JJA has been shown to have maximal cloud responses over forests in *Europe* - one region of the NH Mid-Latitudes. One of the novel aspects of your study is that you're applying your analysis globally, and JJA is not "summer" / summer-like over the whole globe - particularly the tropics where seasonality is quite different, or the SH mid-latitudes where you're effectively looking in winter. There is more land in the NH, so you probably can keep your analysis mostly focused on JJA, but a discussion of "actually we'd expect DJF to probably have a stronger signal in the SH mid-latitudes, and indeed we find that / or we don't find that" would be appropriate. Additionally, wouldn't you expect the largest impact of forests on clouds (or at least on column moisture) over the much of the tropics (esp. the Amazon) to be during the dry / dry-wet transitional season, which is not necessarily JJA (location dependent)?

(See Spracklen et al 2012, doi.org/10.1038/nature11390)

Thanks for these helpful suggestions.

(1) We clarified this point at the beginning of the results and the revised text is shown below:

Here we first focus on boreal summer months (JJA), during which we expect to observe the most pronounced cloud differences between forests and non-forest²² for the majority of the northern hemisphere, while results for other seasons are presented later.

(2) In the revision, we added the cloud effects of other seasons including the local summer (DJF) in the southern hemisphere as well as the dry-wet season in the tropics. The results are consistent with your expectations in DJF. The revised paragraph and the new figure are shown below:

In terms of seasonality, there are notable and region-specific variations in Δ Cloud from both MODIS and MSG data (Figs. 2 and S8). The maximum cloud effects appear in local summer for most areas of the northern hemisphere (JJA) and the mid-latitudes of the southern hemisphere (DJF). In the tropics this occurs during the dry-wet transition, consistent with existing evidence^{16,18,29,30}, cloud inhibition in the Amazon is stronger during the dry season (May to November) than in the wet season, although their timings of maximum effect differ (September for MODIS and July for MSG). Interestingly, cloud inhibition in Central Africa exhibits a larger effect during the dry season in MSG data (predominant JJA) but during the wet season in MODIS data (mixed JJA and MAM). In temperate regions, cloud inhibition in the Southeast US is larger

in summer, while cloud enhancement in Europe shows a slight decline during the snow-free period.

Fig. R6 (Figure 2). Seasonal variations of the potential effects of forests on cloud cover. (a) Months with the maximum magnitude of ΔCloud for MODIS and MSG during the snow-free season. Seasonal variation of ΔCloud is shown in selected regions: (b) Southeast US (97°W to 75°W, 30°N to 40°N), (c) Europe (10°E to 30°E, 47°N to 55°N), (d) Amazon (70°W to 50°W, 16°S to 5°S), and (e) Central Africa (10°E to 33°E, 7.5°S to 0°) for MODIS and MSG. Months with snow cover are shown as the shaded areas.

- Line 93: ie forests have less cloud cover than their non-forest neighboring gridcells? This seems opposite of what I would expect re: the next sentence, when the authors say "due to strong turbulent flux contrast between forest and non-forest in low lats" - can you show the contrast is stronger in low lats than high lats? Wouldn't stronger turbulence = cloud *enhancement* over forests? Some reworking of the text here might clear up my confusion.

Our results in Fig. 1b,d indicated that the strength of both the cloud enhancement and inhibition effects follows a latitudinal dependency with the largest magnitude in the tropical regions and diminished toward higher latitudes. We have revised the explanation for this latitudinal dependency.

It is generally more favorable for convection development at low latitudes because of their high convective available potential energy (CAPE) which decreases at higher latitudes (Riemann-Campe, 2009). The strong convection potential in the tropics can drive a larger cloud effect. More clouds would be developed at low latitudes under favorable conditions over either forest or nonforest patches than those at higher latitudes, and the cloud differences between forest and nonforest will also be greater.

The revised text is shown below:

The strength of these contrasting cloud effects (i.e., cloud enhancement and inhibition) follows a latitudinal dependency with the largest magnitude in the tropical regions and diminished toward higher latitudes (Fig. 1b). This is likely due to preferential conditions for convection development at low latitudes, as indicated by their high convective available potential energy which decreases at higher latitudes²⁷.

- Line 92: “+0.0133” - what is this number? A change in ... cloud frequency? Fractional cloud cover? It is unclear from lines 81-82 what "Cloud" numerically means in this analysis.

The number +0.0133 is for cloud cover fraction which indicates the occurrence frequency of clouds over a period of time. The +0.0133 indicates the global mean magnitude of the cloud enhancement effect. In other words, it means forests with enhanced clouds have on average 0.0133 higher cloud fractions than non-forest. We have revised the text to make it clear:

Most temperate and boreal forests in Eurasia and North America have higher cloud fractions than non-forest, indicating a cloud enhancement effect (positive Δ Cloud) accounting for 63.21% of all grid samples with a global mean magnitude of +0.0133.

- Line 96-97: the fact that there are large enhancements and inhibitions of clouds over forests in the tropics means a single explanation (like is made in the previous sentence" is unlikely to explain the response. It isn't like the cloud response is all one sign and the tropics and all another sign in the high lats, or I'm misreading figure 1b. I don't find figure 1 very supportive of the claims made here.

We apologize for the confusing content and we have clarified this point in the revision. These sentences were to explain the latitudinal pattern of cloud enhancement and inhibition effects which have a larger magnitude at low latitudes. The explanation in the revision is that low latitude regions are generally more favorable for convection development because of their high convective available potential energy (CAPE) which decreases at higher latitude (as we responded to earlier comment). Therefore, more clouds would be developed at low latitudes under favorable conditions over either forest or nonforest patches than those at higher latitudes, and the cloud differences between forest and nonforest will also be greater.

- Line 102-103: “inhibition stronger in central Africa, weaker in the Amazon regions...” Why? And what about the strong mix of positive and negative signals indicated in figure 1?

The estimated cloud effects can differ in magnitude and even in sign between different cloud datasets. Since cloud cover datasets from different satellites have their own strengths, limitations, and uncertainties, we used multiple cloud datasets to estimate cloud effects to reduce the data uncertainty. There are indeed differences in the cloud effects in central Africa where MSG data showed strong inhibition effects while MODIS data showed mixed positive and negative cloud effects. Additionally, the cloud effect estimated from Sentinel-5P pointed out cloud inhibition in

central Africa (Fig. R7), though we did not report Sentinel-5P results in the main text due to its short time period and different cloud amounts compared to MODIS and MSG data. The specific reason for the different patterns among data sources is currently unknown but we do agree that cloud data present a key source of uncertainty in our results, which is worth further discussion.

In the revision, this sentence has been revised to be more specific:

At 14:00 local time, cloud inhibition is stronger in central Africa while weaker in the Amazon regions compared to MODIS data (Fig. 1c,d).

Fig. R7 (Figure S13). The potential effect of forests on cloud cover (ΔCloud) based on Sentinel-5P data and (b) the percentage of negative ΔCloud in four selected regions from Sentinel-5P, MODIS, and MSG cloud cover data. The four black rectangles in panel (a) denote four hotspots regions, Southeast US (97°W to 75°W, 30°N to 40°N), Amazon (70°W to 50°W, 16°S to 5°S), Central Africa (10°E to 33°E, 7.5°S to 0°) and Europe (10°E to 80°E, 47°N to 65°N). Note that the Europe box here is larger than that in Fig. 2. The dashed black horizontal line in panel (b) represents the 50% percent line, with value >50% indicating more cloud inhibition and <50% indicating more cloud enhancement of forests.

- Line 116-117: “cloud inhibition is stronger during the dry season in the Amazon, but amplified during the wet season in Central Africa” - why? without presenting some potential explanations for why two tropical forests at the same latitude would produce opposite cloud responses, these correlations don't provide particularly compelling evidence for the cloud responses being driven by the forests. I'm not saying they're not - they very well could be, and the seasonality and climatology of the Amazon vs. central African rainforests have

some distinct differences. So, it would strengthen the paper to discuss how those differences could physically be expected to produce different responses of clouds to forest change, and then discuss if your results do/do not support those physical mechanisms, would be much more compelling.

We conducted further analysis and found that the different seasonalities of cloud effects for tropical forests in Amazon and Central Africa are likely due to cloud data. In the Amazon, both MODIS and MSG indicated stronger cloud inhibition in the dry season although their timings of the maximum effect differ (September for MODIS and July for MSG). However, MODIS and MSG showed different seasonalities in central Africa where the cloud inhibition exhibited a larger effect during the dry season with MSG data (predominant JJA) but wet season with MODIS data (mixed JJA and MAM).

The above new analyses and a new figure have been added to the revised manuscript:

In terms of seasonality, there are notable and region-specific variations in ΔCloud from both MODIS and MSG data (Figs. 2, S8). The maximum cloud effects appear in local summer for most areas of the northern hemisphere (JJA) and the mid-latitudes of the southern hemisphere (DJF). In the tropics this occurs during the dry-wet transition, consistent with existing evidence^{16,18,29,30}, cloud inhibition in the Amazon is stronger during the dry season (May to November) than in the wet season, although their timing of maximum effects differs (September for MODIS and July for MSG). Interestingly, cloud inhibition in Central Africa exhibits a larger effect during the dry season in MSG data (predominant JJA) but during the wet season in MODIS data (mixed JJA and MAM). In temperate regions, cloud inhibition in the Southeast US is larger in summer, while cloud enhancement in Europe shows a slight decline during the snow-free period.

Fig. R6 (Figure 2). Seasonal variations of the potential effects of forests on cloud cover. (a) Months with the maximum magnitude of ΔCloud for MODIS and MSG during the snow-free season. Seasonal variation of ΔCloud is shown in selected regions: (b) Southeast US (97°W to 75°W , 30°N to 40°N), (c) Europe (10°E to 30°E , 47°N to 55°N), (d) Amazon (70°W to 50°W , 16°S to 5°S), and (e) Central Africa (10°E to 33°E , 7.5°S to 0°) for MODIS and MSG. Months with snow cover are shown as the shaded areas.

The specific reason for the inconsistency among data sources is currently unknown but we do agree that cloud data present a key source of uncertainty in our results and worth further discussion. Under this circumstance, it is important to have independent evidence from other studies to verify the seasonal pattern. For the Amazon, there is ample observational evidence from other research to corroborate the estimated cloud effects and their seasonality. However, such evidence for central Africa is lacking. Thus, further research is needed in these understudied regions to resolve this issue.

We added a discussion on this matter in the revision:

However, it is worth noting that the estimated cloud effects, despite their broad agreement in the global pattern across datasets, can differ in magnitude and even in sign in certain regions (e.g., the inconsistent cloud effects and their seasonality in central Africa), suggesting cloud data to be

a key uncertainty source for our analysis. This is exacerbated by the lack of direct observations in regions like central Africa which have received little attention compared to the Amazon, hindering comparisons against other available evidence. This highlights the importance of dedicated observational efforts in specific regions, especially those understudied, to provide complementary information to our global-scale analysis.

- Figure 2: MSG and MODIS don't appear to agree on the attribution here. MSG is largely blue across central Africa, MODIS is largely red/rainbow. Please explain this disagreement.

This is because the attribution analysis was based on the estimated cloud effects of MODIS and MSG. Therefore, the inconsistency appearing in the cloud effects (e.g., central Africa) across datasets would also appear in the attribution analysis. As for this inconsistency, please refer to our previous reply.

- Lines 126-127: Would it not be a more accurate representation of ref. 27 to say that "human-causes forest loss tends to be stronger at lower elevations and in regions of non-complex terrain"? Plenty of forests are located in regions of simple terrain and low elevation, and plenty more would be there if people didn't cut them down. The way this is written right now is just misleading in that it makes it seem like forests preferentially grown on steep mountain slopes, while actually they would happily grow many other places, they're just easier to cut down in many of those other places.

Thanks for this suggestion. Human impact indeed caused this tendency of forest distribution. We revised this sentence to reflect this important point:

The estimated cloud effects of forests could be confounded by orographic clouds because of the dual influences of topography on forest distribution and cloud formation. Human impacts resulted in a global tendency of existing forests to be located at more complex terrain with a higher elevation than non-forest²⁹ .

- Line 129 & figure S7: Some clarification here is needed, I think - to explicitly state that orography (when moist air is pushed up the slope, anyhow) promotes cloud formation, and the fact that many forests are left in regions of complex topography might then result in the forests being under clouds simply because they are on a slope, and not because they are a forest... I think that is what the authors are trying to say here, it just wasn't super clear. Similarly, I was confused looking at figure S7 what it was trying to convey. I think the idea was supposed to be that the forest on the hill has clouds because it is on a hill, not because it is a forest. In that case, maybe replacing the forest in (b) with grass would be more helpful, and even adding 2 more panels where the hill is flattened and with a schematic of how the forest would impact clouds *in the absence* of a hill might be helpful.

Thanks for the suggested changes. That is exactly what we want to say about this figure: vegetation located on slope areas would have more clouds than those located on flat areas simply because of the orographic effect, regardless of vegetation types.

(1) In the revision, the sentence has been revised to clarify this point:

The high elevation *per se* could facilitate cloud formation through the orographic lifting of moist air³⁰, leading to increased cloud cover over the forest located at a higher altitude (Fig. S5).

(2) Following the reviewer's suggestion, we modified the schematic figure in panel b by replacing forest with grass to better demonstrate the orographic effect, (see revised figure below, Fig. R8). As for the additional two panels where the fill is flattened (no orographic effect), they have already been shown in the main text as Fig. 4.

Fig. R8 (Figure S5). Schematic of orographic clouds which could potentially confound the forest effects on cloud cover. (a) Orographic induced enhanced cloud cover and (b) inhibited cloud cover over forests.

- Figure 3b: What are the circles for? (add to caption)

The three circles denote the locations of cloud inhibition that correspond to negative ΔH in Amazon, Central Africa, and Southeast US.

We added description below to the figure caption:

The three circles marked in (a) and (b) denote the locations of cloud inhibition which correspond to the negative ΔH in Amazon, Central Africa, and Southeast US.

- Figure 3c: What are the lines? The paired sites? If so, they're pretty far away. If not, what are they? Why do only some sites have lines? The caption says the lines indicate flux tower clusters, but ... is a cluster the same as a paired site, or is it a place where there are lots of paired sites?

Thanks. The description has been added to the revised caption:

The connection lines with a dot in panel (c) indicate the location of flux tower clusters where multiple flux pairs are close in distance.

- Figure 3d: would this be considered a statistically significant slope? I'm more familiar with the Pearson correlation test (I'm not at all saying it is better/worse in this case), but doing a quick search of the Spearman's correlation test doesn't tell me how its p-value is

calculated. Is it the p-value from a student's t-test? If so, please add that to the caption to explain. If not, what is it?

The Spearman's correlation is a rank correlation that was calculated using the `spearmanr` function of the `scipy.stats` module in Python (<https://docs.scipy.org/doc/scipy/reference/generated/scipy.stats.spearmanr.html>). By checking the documentation and code of this function, the p-value is calculated from a two-tailed t-test.

We added an explanation about spearman's correlation coefficient and its p-value to the caption: The spearman's correlation coefficient (\$\rho\$ ), which is a nonparametric measure of rank correlation, is calculated by the `spearmanr` function of `scipy.stats` module in Python with its p-value (\$p\$ ) determined by a two-tail t-test.

- Figure 3: What is "potential effect of forest on sensible heat"? The paper would benefit from more clearly laying out the definitions for the metrics the authors use.

The potential effect of forests on sensible heat originally refers to the sensible heat difference between forest and non-forest. In the revision, the "potential effect of forest on sensible heat" is no longer used and is replaced by the sensible heat differences between forest and nonforest to be more specific.

- Line 141: "One pair in the Amazon is not shown on the map" - why?

This is because there is only one site pair in the Amazon region. Displaying the single site pair would require a separate map figure of the entire Amazon but it actually gives very little information. For this reason, we only show key regions on the map with several site pairs. The information of all site pairs can be found in Table S1.

- Line 150-151: "forest has ... lower land surface temperature, increased evaporation and soil moisture than nonforest vegetation". Not always. The authors could hedge here and say "generally...". But Some agricultural areas in particular have higher fluxes than forests, and seasonally forests can have really low ET (e.g. in the spring in the high latitudes when the deeper soils are frozen) compared to grasses/shrubs with shallow root systems. However, I don't think the authors need to get into a detailed deep-dive on nuanced forest vs non-forest flux responses under specific conditions, seasons, etc. – rather, they could simply soften the statement they make a bit so as not to distract readers like me who will see and immediately ask "but what about this special case?"

Thanks for this suggestion and we fully agree with the reviewer. As correctly pointed out by the reviewer, there are always exceptions for those described biophysical differences between forest and nonforest. In the revision, the sentence is revised to soften the tone:

In terms of biophysical differences, forests generally have reduced albedo, higher roughness, lower land surface temperature (LST), increased evapotranspiration and soil moisture compared to non-forest vegetation^{4,5}.

- Lines 155-158: Figure 3c is just from the paired flux towers, isn't it? Is the slope robust if you use CLM and satellite observations, too? It seems like 3c should be showing the regression for all methods used, because 3b is just showing the change in H, not how the change in H correlates with (or doesn't) the in clouds. change It is unclear what b means

by "satellite" (MODIS? MODIS and MSG?), but if it is MODIS, the reader could try and mentally regress delta H from b to delta cloud in a, but delta cloud is never shown for CLM.

(1) Since the accurate estimation of sensible heat is still a difficult and challenging task, we used three independent datasets to estimate ΔH between forest and nonforest (including CLM, satellite estimates, and flux towers), and their relationships with ΔCloud are presented in different ways. This choice is made because of the uncertainty and limitation of these sensible heat datasets. For satellite estimates, sensible heat is estimated as the residual energy flux ($H + G = \text{net radiation} - \text{LE}$) which includes ground flux as well. For CLM estimates, land surface models are known to have biases in simulating H and ET . Therefore, large uncertainties can be expected for both the satellite and CLM estimates of H , as indicated by their differences in the spatial pattern shown in Fig. 3a. For these reasons, we did not expect strong quantitative relationships between ΔH from satellite/CLM and ΔCloud . Instead, their relationships with ΔCloud are presented as the broad spatial correspondence between cloud inhibition and negative ΔCloud (highlighted by the three circles on the map) or cloud enhancement and positive ΔCloud in a qualitative way. Although flux tower measurements have their own problems (e.g., few sites, sites distance), they are often considered to be the "ground truth" to validate satellite and model estimates, and are thus considered to be the most accurate method among the three datasets. Therefore, we only present the quantitative relationship (scatterplot) between ΔH from paired flux towers and the corresponding MODIS ΔCloud in Figure 3c.

As requested by the reviewer, here in the response letter we provide scatterplots for ΔH from satellite/CLM against ΔCloud (Fig. R4). Results support the positive relationships between these two with spearman correlation coefficients of 0.14 ($p < 0.01$) and 0.11 ($p < 0.01$) for satellite- and CLM-derived ΔH , respectively. Although the relationships are not as strong as those for flux towers (possibly due to large uncertainties in the H estimates), they confirm the positive linkage between ΔH and ΔCloud shown in Fig. 3b.

This information has been added in the following sentence:

Such a spatial co-occurrence is further confirmed by the positive relationship between observed ΔH from paired flux sites and MODIS ΔCloud ($\rho = 0.34$, Fig. 3d), and those between ΔH derived from satellite/CLM and ΔCloud (weaker but statistically significant; not shown), suggesting that cloud enhancement is more likely to occur when sensible heat over the forest is larger than nearby non-forest, and cloud inhibition occurs when sensible heat over the forest is smaller.

Fig. R4. The regression between ΔCloud and ΔH from (a) satellite and (b) CLM. The correlation coefficient refers to the spearman correlation coefficient.

(2) The "Satellite" in Fig. 3b means " ΔH based on satellite estimate", as explained in the figure caption: The sensible heat difference between forest and non-forest (\$\Delta\text{H}\$ ) estimated from satellite data⁴. It is also explained in the method section as:

Satellite estimates provide changes in the combined sensible heat and ground heat fluxes (H+G) under different land cover conversions at 1° spatial resolution based on MODIS data (a total of 45 pairs of land conversions for "HG_IGBPdet").

- Lines 158-160: I'm pretty confused what is being shown in 3b. Did you take forested gridcells and nearby non forested gridcells and difference the H between those?

Yes, that is correct. In short, the sensible heat differences between forest and nonforest for satellite are extracted by 1) extracting the specific forest and nonforest types from moving windows and then aggregated at each one-degree box (e.g., "evergreen broadleaf" for forest type and "crop" for nonforest type) 2) and then extracting the corresponding sensible heat change value at the same grid box from the 45 pairs of land conversions defined by the satellite dataset of Duveiller 2018.

For more details please refer to the description below from our method section:

Satellite estimates provide changes in the combined sensible heat and ground heat fluxes (H+G) under different land cover conversions at 1° spatial resolution based on MODIS data (a total of 45 pairs of land conversions for "HG_IGBPdet"). The combined fluxes of H+G were estimated as the residual of surface energy components as described in Ref⁴. Due to the small contribution of G to H+G, we referred to "H+G" as "H" for simplicity in the following text and the main text. To obtain sensible heat differences between forest and non-forest (\$\Delta\text{H}\$ ) that are compatible with \$\Delta\text{Cloud}\$, we extracted the dominant land cover type for unchanged forest (e.g., evergreen broadleaf) and non-forest pixels (e.g., crop) within each moving window from the ESA land cover product. The dominant land cover types for forest and non-forest were upscaled to 1° resolution with the "major" method (figure not shown for 1°, but a similar one for 0.5° is shown in Fig. S9). For each one-degree grid box with a dominant forest type (e.g., evergreen broadleaf) and non-forest type (e.g., crop), \$\Delta\text{H}\$ can be extracted from the corresponding sensible heat change value that matches the specific land conversion (e.g., evergreen broadleaf to crop) at the same grid box from the 45 pairs of land cover conversions defined within the "HG_IGBPdet" dataset.

- Lines 167-171: I'm confused at what the authors are getting at here; this seems sort of like saying what the Bowen ratio is twice. It isn't because the Bowen ratio is small that LH is comparatively large and SH is comparatively small. That is the definition of the Bowen ratio. So, LH is big and SH is small therefore the Bowen ratio is small seems redundant... I don't understand what they're getting at.

Thank you for your comment. This sentence has been changed to:

Forests at low latitudes under humid climates have smaller Bowen ratios as most available energy goes into latent heat rather than sensible heat, resulting in even smaller sensible heat compared to non-forest, while forests at higher latitudes under drier climates have larger Bowen ratios leading to the opposite effect.

- Line 170: This is the closest the authors get so far to saying anything about how vegetation influences boundary layer development and the lifting of moist parcels of air to the LCL. Which seems like a critical part of the physical argument. The authors should consider devoting more space to this topic.

Thanks for this suggestion. We expanded this point in the next paragraph as:

Enhanced sensible heating, which typically occurs over the forest relative to non-forest vegetation³⁴, serves as a major lifting mechanism to initiate convection and the growth of boundary layer^{29,32}, and formation of a "forest-breeze" analogous to a sea-breeze. Specifically, the high sensible heat elevates the atmospheric boundary layer (ABL) such that the lifting condensation level (LCL) is lower than the ABL depth^{18,29,39}, thereby supporting low-level cloud formation.

- Lines 179-195: This is a clear and nice summary, though is more of a lit review than a "results from this study". Maybe the authors could weave in what their results add to the state-of-the-science here. In addition, including a reference/discussion of how the authors' results relate to the literature exploring the response of clouds and precipitation to soil moisture regimes would be beneficial here, as it is related to the argument as they show in figure 4 (except being driven by sensible/latent heat partitioning as a function of soil moisture).

Possibly useful references from that section of the literature:

Ek, M. B., & Holtslag, A. A. M. (2004). Influence of soil moisture on boundary layer cloud development. *Journal of Hydrometeorology*, 5(1), 86–99. [https://doi.org/10.1175/1525-7541\(2004\)005<0086:IOSMOB>2.0.CO;2](https://doi.org/10.1175/1525-7541(2004)005<0086:IOSMOB>2.0.CO;2)

Welty, J., Stillman, S., Zeng, X., & Santanello, J. (2020). Increased Likelihood of Appreciable Afternoon Rainfall Over Wetter or Drier Soils Dependent Upon Atmospheric Dynamic Influence. *Geophysical Research Letters*, 47(11), 1–9. <https://doi.org/10.1029/2020GL087779>

Guillod, B. P., Orlowsky, B., Miralles, D. G., Teuling, A. J., & Seneviratne, S. I. (2015). Reconciling spatial and temporal soil moisture effects on afternoon rainfall. *Nature Communications*, 6(March), 1–6. <https://doi.org/10.1038/ncomms7443>

Koster, R., Dirmeyer, P. A., Guo, Z., Bonan, G., Chan, E., Cox, P., ... Vasic, R. (2004). Regions of Strong Coupling Between Soil Moisture and Precipitation. *Science*, 305, 1138–1140.

Dirmeyer, P. A. (2011). The terrestrial segment of soil moisture-climate coupling. *Geophysical Research Letters*, 38(16), 1–5. <https://doi.org/10.1029/2011GL048268>

Koster, R. D., Guo, Z., Dirmeyer, P. A., Bonan, G., Chan, E., Cox, P., ... Yamada, T. (2006). GLACE: The Global Land-Atmosphere Coupling Experiment. Part I: Overview. *Journal of Hydrometeorology*, 7(4), 590–610. <https://doi.org/10.1175/JHM510.1>

Thanks a lot for these useful references which are closely related to our topic. In the revision, we have read through all of them and added the following content to bridge our results to the literature on soil moisture and precipitation.

The mesoscale circulation mechanisms associated with the inhibited clouds over forests echo a large body of research in land-atmosphere interactions emphasizing the critical role of surface fluxes and soil moisture anomalies in the atmospheric boundary layer, clouds, and precipitation processes^{45,46}. The preference of convective clouds and precipitation over drier soils identified in previous studies using observations^{47,48}, analytic models⁴⁹, and numerical simulations⁵⁰ is in line with the enhanced clouds over regions of higher sensible heat flux, as we show here for forest and non-forest transitions. Interestingly, the preference of clouds over non-forest found in Southeast US, Amazon, and Central Africa of our study also roughly correspond to those reported to exhibit a temporal preference of afternoon rain with drier soil conditions (Central US, West Amazon, and parts of the Sahel and Equatorial Africa)⁴⁸.

- Figure 5: These trends, except maybe over the Maritime Continent, are very small (and for the Maritime Continent, showing it is forest loss and not ocean variability that is driving the signal would be necessary). Could the authors give some context to the magnitude of the changes and slopes they're showing here?

Similar to the potential effect of forests, the forest loss impact is estimated by comparing the trends of cloud cover fraction between forest loss locations with nearby undisturbed forests. The signal of natural variability in cloud cover would be removed through the subtraction by assuming cloud changes driven by atmosphere/ocean variability are the same for forest loss and nearby unchanged forest.

The numerically small trend values are due to the narrow range of cloud cover fraction (0~1). In the revision, we added statistical significance of those trends (p-value to measure how significant forest loss induced cloud cover trend against cloud variability) on the figure and in the revised text.

Fig. R9 (Figure 5). Impacts of forest loss on JJA cloud cover based on MODIS data from 2002 to 2018. (a) The accumulated forest loss fraction from 2001 to 2018. (b) The actual cloud impact of forest loss ($\Delta\text{Cloud}_{\text{loss}}$), defined as the mean cloud difference between forest loss location and nearby unchanged forests from 2002 to 2018. Four hotspot regions that experienced intensive forest loss are highlighted in panels (c) to (r), showing their forest loss fractions, mean $\Delta\text{Cloud}_{\text{loss}}$ during the study period, and regional and temporal trends of $\Delta\text{Cloud}_{\text{loss}}$ between 2002 and 2018 (column-wise). The cloud impacts in forest loss hotspot regions are estimated from grid boxes with tree cover loss fraction > 0.05. The green dashed line in the last column (f,j,n,r) shows the

annual tree cover difference between forest loss location and nearby unchanged forests (ΔTree) and the red solid line shows the temporal trends of $\Delta\text{Cloud}_{\text{loss}}$ (unit: %/year).

Please see the revised texts with p-value:

Furthermore, cloud enhancement in these forest loss hotspots became increasingly stronger with the decline and fragmentation of tree cover⁵³, which translates into total cloud fraction increases of 0.78% (0.046%/year, $p < 0.01$), 1.19% (0.070%/year, $p < 0.01$), and 0.09% (0.005%/year, $p = 0.40$) throughout 2002 to 2018 for Amazon, Indonesia, and Southeast US hotspots, respectively (3rd and 4th column in Fig. 5).

The cloud reduction also exhibited a strengthening trend, resulting in a total reduction in cloud cover fraction of -0.20% (-0.012%/year, $p = 0.16$) from 2002 to 2018 (Fig. 5q, r).

The statistical significance tests showed that the $\Delta\text{Cloud}_{\text{loss}}$ trends are significant at 0.05 level for Amazon and Indonesia (the maritime continent) hotspots, but not in southeast US ($p = 0.40$) and Siberia ($p = 0.16$) hotspots. The insignificant trends probably reflect the fact that the magnitude of

cloud impact generally declines toward higher latitude, while the cloud cover variability increases instead (Fig. R1). This results in a decreasing signal-to-noise ratio toward higher latitude, making the cloud impact insignificant compared to the relatively large cloud cover variability at high latitudes. Since the trends are used to estimate the accumulated cloud cover change due to forest loss, statistical significance does not affect the estimation given the insignificant trends are also very small.

A discussion on this matter has been added:

However, due to the decreasing magnitude of cloud effect and increasing cloud interannual variability toward higher latitudes (Fig.1, Fig. S1), the signal-to-noise ratio of the cloud effect also decreases at higher latitudes, making forest loss induced cloud cover changes there less detectable than those occurring at lower latitudes.

Fig. R1 (Figure S1). (a) The spatial distribution of multi-year mean MODIS JJA cloud cover fractions at 0.05 ° from 2002 to 2018. (inset: MSG JJA cloud cover fractions from 2004 to 2013). (b) The standard deviation of the time series of MODIS JJA cloud cover fraction during 2002-2018 and (c) its latitudinal pattern.

- Lines 229-231 (and figure 5): some indication of statistical significance is necessary, because these relationships are extremely small.

Statistical significance for the trends of $\Delta\text{Cloud}_{\text{loss}}$ has been added, please refer to our response to the previous comment.

- Line 232: I don't think figure 5f shows this. 5f just shows the study period, or I'm missing something. Please clarify.

Fig. 5f shows ΔCloud (red line) and ΔTree (green dashed line) of each year during the study period. In 2002, the forest loss location in Amazon already had lower tree cover ($\Delta\text{Tree} < -0.1$) than unchanged forest, due to the forest loss legacy that happened before 2001. This forest loss legacy led to higher cloud cover (positive ΔCloud) over forest loss locations at the beginning of the study period.

This sentence has been revised to:

Note that in the Amazon, forest loss legacy before 2001 had already caused lower tree cover ($\Delta\text{Tree} < -0.1$) in 2002, which is responsible for the enhanced cloud cover (positive $\Delta\text{Cloud}_{\text{loss}}$) observed over the forest loss locations at the beginning of the study period (Fig. 5f).

- Can the authors comment on the biases in their ET dataset?

The ET dataset (MOD16A2 Version 6) is a 8-day composite product (500 m) that was derived from the Penman-Monteith equation with high simulation accuracy. The MOD16A2 product has been evaluated in regions such as Asia (Kim et al., 2012; Faisal et al., 2021), South America (Degano et al., 2021), and US (Ruhoff et al., 2012), and over various land covers including forest, agriculture, grass, and shrubland etc. These validation efforts reported an overall good performance of MOD16A2. However, we acknowledge that the ET dataset is not perfect with known local biases. But for our global-scale study, we feel that the quality of MOD16A2 is sufficient given that ET data are only used for a supplementary purpose.

- Figure S4: the color maps in both (a) and (b) are extremely hard to me to tell apart. I look at (a) and just see "orange-ish everywhere", rather than a clear "altostratus" vs "stratocumulus" vs "others". Similarly in (b) the bars are just a bit hard on the eyes. Even adding a black outline to each bar here would probably help a lot.

Thanks for the comment. For panel (a), we adjusted legend colors on the map by setting non-convective cloud types to grey to highlight the pattern of the three convective cloud types. The percentage of all nine cloud types can be seen from panel (b). The revised figure is shown below:

Fig. R10 (Figure S7). (a) The dominant cloud type for the cloud effects of forests in JJA based on Sentinel-5P. Note that only the three convective cloud types are shown in color. (b) The percentage of each dominant cloud type globally and in four selected regions: Southeast US (97°W to 75°W, 30°N to 40°N), Amazon (70°W to 50°W, 16°S to 5°S), Central Africa (10°E to 33°E, 7.5°S to 0°) and Europe (10°E to 80°E, 47°N to 65°N). Note that the Europe box here is larger than that in Fig. 2.

- Figure S5: what *area* of each region (as a weight - e.g. 10%, or 80% etc) show positive vs negative effects? Including such weighting in this figure, and in the figures in the main text, on the line plots of both positive and negative effects would help the reader know if there are equal and opposite magnitudes of positive and negative response, or if the areas of positive response are much smaller than the areas of negative response (or vice versa)

Thanks for this good suggestion. We added the percentage of positive and negative Δ Cloud in Fig. 1. The revised figure is shown below:

Fig. R11 (Figure 1). The potential effects of forests on June-August (JJA) cloud cover fraction and their attribution. The potential effect is defined as the differences in cloud cover fraction between forests and nearby non-forest (ΔCloud) from MODIS and MSG satellites that detect clouds. (a) Potential effects of forests on cloud cover fraction based on MODIS data from 2002 to 2018 (overpass at 13:30 local time) and (b) their latitudinal patterns with cloud enhancement and inhibition effects separated. (c,d) Potential effects of forests on cloud cover fraction based on hourly MSG data from 2004 to 2013 (overpass at 14:00 local time) and (e) the timing of the maximum effect during a day. The numbers in Panels (b) and (d) show the percentage of cloud enhancement (red) and inhibition (blue). (f) Attribution of cloud effects of forests to tree cover and elevation based on MODIS and MSG data. The five attribution categories include tree cover induced cloud increase (Tree+) and decrease (Tree-), orography induced cloud increase (Orography+) and decrease (Orography-), and other unexplained effects. The percentage of each attribution category is calculated based on the MODIS results.

- Figures S2, S3, S8, and generally throughout the paper – please make it clear what “delta cloud” is (e.g. change in cloud fraction)

In the revision, we added clarification in their figure captions to specify Δ Cloud as "the effects of forests on cloud cover fraction".

Minor comments:

- **Line 35-36: this usage of brackets is confusing. Could write the sentence only slightly longer but much more clearly as:**

"driven by sensible heating where cloud enhancement is more likely to occur when sensible heat over the forest is larger than nearby non-forested regions, and cloud inhibition occurs when sensible heat over the forest is smaller."

See the below EOS article from 2010 on the subject – I'm not alone in finding this confusing!

<https://eos.org/opinions/parentheses-are-not-for-references-and-clarification-saving-space>

We really appreciate these helpful guidelines on the usage of brackets and we fully agree with it. The sentence has been revised to:

The spatial variation in the sign of cloud effects is driven by sensible heating where cloud enhancement is more likely to occur over forests with larger sensible heat, and cloud inhibition over forests with smaller sensible heat.

- **Line 37: "opposite cloud cover changes" – opposite of what?**

It means "cloud cover changes in opposite directions". This sentence has been revised to clarify this point :

Ongoing forest cover loss has led to cloud increase over forest loss hotspots in the Amazon (+0.78%), Indonesia (+1.19%), and Southeast US (+0.09%), but cloud reduction in East Siberia (-0.20%) from 2002-2018.

- **Line 43: "processese" – do the authors mean processes?**

Thanks for pointing out the typo and it has been corrected.

- **Line 62 – "inhibited clouds over SOME forest"**

Thank you for this suggestion which makes the statement more accurate. The revised sentence is:

These results revealed inhibited clouds over some forests (e.g., West Africa²⁰) at a realistic scale which seemingly contradicts the highly hypothetical GCM results²¹ and the enhanced cloud observations over forests in other regions (e.g., western Europe²² and Central America²³).

- **Figure 1 - I would suggest the authors reverse their color bar here, such that blue = more clouds and red = less clouds. Intuitively, red suggests "dry" and blue suggests "wet", and since clouds are a moisture related field this could be helpful.**

Thanks for this suggestion. We did not use this colormap for reasons below. The common usage of the contrasting blue-red colormap is red for high values and blue for low values. If we use the

reversed colors to represent cloud differences (i.e., red for negative and blue for positive ΔCloud , Fig1 and 3a), this will contradict the color scheme for sensible heat differences (ΔH) in Figure 3b (red for positive and blue for negative ΔH) and creates inconsistency. For Figure 3b, what we want to highlight is the spatial agreement between ΔCloud and ΔH , therefore, it is important to use the same color scheme for both to better highlight their spatial correspondence. For this consideration, we decided to use the red and blue colors to represent positive and negative values consistently for both ΔCloud and ΔH .

- Figure 1e – again, the authors might want to consider a different set of 4 colors. It is extremely hard for my eyes to tell the difference between the cyan and green colors used here.

Thank you. In the revision, we replaced cyan with yellow to improve readability of the 4 colors. Please check the revised Figure 1 (Fig. R11).

- Line 77: a brief introduction to MSG and MODIS is necessary since the methods appear at the end of papers in these formats. Just "satellites that detect clouds" is good enough.

Thanks for this suggestion.

(1) In the revision, we first added an explanation of MODIS and MSG in the figure caption:
The potential effect is defined as the differences in cloud cover fraction between forests and nearby non-forest (ΔCloud) from MODIS and MSG satellites that detect clouds.

(2) At the beginning of the results section, we added description below to help readers understand the data sources we used to quantify the cloud effects.
Here, the cloud effect is measured by cloud cover fraction derived from Moderate Resolution Imaging Spectroradiometer (MODIS) and Meteosat Second Generation (MSG) satellite data, which represent the occurrence frequency of clouds over a period of time with a valid range of 0-1 (Fig. S1).

- Line 82: why is (spatial) in brackets?

It was used to emphasize "spatial" comparison rather than temporal comparison. The brackets were removed in the revision.

- Line 99: "splited" -> "split"

Thanks for pointing out the error. It has been corrected.

- Line 101: necessary to say something about MSG getting hourly resolution but non-global coverage (could say this when you briefly introduce MODIS and MSG)

We added a brief description of MSG as:

Similar spatial and latitudinal patterns can also be seen from geostationary MSG satellite data with high temporal resolution (i.e., hourly) but non-global coverage.

- Line 143: So the cloud fraction for the tower sites was taken from MODIS? (Just a clarification question here)

Yes, it is. The ΔCloud at tower site pairs were extracted only from MODIS because of its global coverage.

- Line 157 (ref 31) - is this the citation the authors meant to use? I thought this study looked at accounting for biomass heat storage on diurnal temperatures... could the authors elaborate on why that is the appropriate CLM simul like in the satellite observations?) Why not use a CLM simulation with deforestation?

(1) Ref 31 was used in the initial submission as it contains a description of the CLM simulation. Here we realize that the ref might not be appropriate for the CLM model. Therefore, here we changed the reference to the CLM documentation (ref 60) and used ref 31 for model simulation elsewhere:

Ref 33. Lawrence, D. et al. Technical description of version 5.0 of the Community Land Model (CLM). NCAR/TN-478+STR NCAR Tech. Note (2018).

The years 1997 to 2001 were the spinup period and excluded from the analysis (please see detailed description in Ref 72).

(2) In the method section, we added an explanation for why offline CLM simulation is an appropriate method to compare against satellite estimates:

In CLM, different types of vegetation within a grid cell are represented as separated tiles of different plant functional types (PFTs). We used subgrid PFT-level model outputs to calculate sensible heat differences between different land cover types within the same model grid. The subgrid tiles within a model grid cell share the same atmospheric forcing, therefore replicating the assumption of similar meteorological conditions of the space-for-time approach¹² .

We did not use the typical deforestation experiment, because the subgrid capability of CLM enables a more flexible way to extract the differences in biophysical properties between different land cover types all at once.

- Line 164-165: bracket usagation to explore here, and how exactly they're calculating delta H (using a similar forest vs nonforest site nearby

Thank you.

(1) We have revised this sentence to avoid the inappropriate usage of brackets for saving space: Such a spatial co-occurrence is further confirmed by the positive relationship between observed ΔH from paired flux sites and MODIS ΔCloud ($\rho=0.34$, Fig. 3d), and those between ΔH derived from satellite/CLM and ΔCloud (weaker but statistically significant; not shown), suggesting that cloud enhancement is more likely to occur when sensible heat over the forest is larger than nearby non-forest, and cloud inhibition occurs when sensible heat over the forest is smaller.

(2) In the method section, we clarified the calculation of ΔH between forest and nonforest flux tower sites:

ΔH was calculated as the mean sensible heat flux difference between the paired forest and non-forest site during the daytime (8:00 to 16:00).

- Line 167: "levels!" -> "levels" (typo)

Thanks! The typo has been corrected.

- Line 192: brackets

We have removed all use cases of brackets for saving space. The brackets here are for clarification and explanation purposes.

Reviewer #2 (Remarks to the Author):

The authors present an investigation of cloud cover differences between forests and nearby unforested areas and find regionally different behavior of cloud enhancement and inhibition.

In general, the topic is important and of interest to the community, given the fact that changes in forest cover are likely accompanied by changes in cloud cover and potentially precipitation. In the light of global change a better understanding of the direction of these processes and the underlying mechanisms would be appreciated.

I have several major comments that I think are important to address before publication.

We thank this reviewer for commending the importance of our study, and we have made significant efforts to address these comments in the revision.

1. Proposed mechanisms for cloud cover change and confidence.

a) The authors present a mechanism based on differences in sensible heat flux. At the same time, figure 3b does not show very strong relationship as indicated by the p-value and the fact that most sites seem to show a Δ_{cloud} in the range of 0-0.02 irregardless of Δ_{H} . Also, Figure S12 shows that MODIS and Sentinel do not agree on the sign of the effect in 2 of the 4 regions. It appears that there is a clear signal in the Amazon and in EurAsia, but not in the US and Central-Africa. I feel that caution should be taken in explaining cloud cover differences based on a mechanism if we cannot establish the overall sign of the effect for a given region depending on the data used.

(1) Since the accurate estimation of sensible heat is still a difficult and challenging task, we used three independent datasets to estimate ΔH between forest and nonforest (including CLM, satellite estimates, and flux towers) and each of these datasets has its own uncertainty and limitation. For satellite estimates, sensible heat is estimated as the residual energy flux ($H + G = \text{net radiation} - LE$) which includes ground flux as well. For CLM estimates, land surface models are known to have biases in simulating H and ET . Large uncertainties can be expected for both the satellite and CLM estimates of H , as indicated by their differences in the spatial pattern as shown in Fig. 3b. And we did not expect strong quantitative relationships between ΔH from satellite/CLM and Δ_{Cloud} . Instead, their relationships with Δ_{Cloud} are presented as the broadly spatial correspondence between the locations of cloud inhibition and negative Δ_{Cloud} (highlighted by the three circles on the map) or cloud enhancement and positive Δ_{Cloud} .

Although flux tower measurements have their own problems (e.g., few sites, distance between sites), they are considered to be the most accurate method for estimating sensible heat among the three datasets, and are often used as the "ground truth" to validate satellite and model estimates. Therefore, we only present the quantitative relationship (scatterplot) between ΔH from paired flux towers and the corresponding Δ_{Cloud} from MODIS in Figure 3c. The numerically small values of Δ_{Cloud} are due to the narrow range of cloud cover fraction (0~1).

In the response letter, as also requested by the 1st reviewer, here we provide scatterplots for ΔH from satellite/CLM against Δ_{Cloud} (Fig. R4). Results support the positive relationships between these two with spearman correlation coefficients of 0.14 ($p < 0.01$) and 0.11 ($p < 0.01$) for satellite-

and CLM-derived ΔH , respectively. Although the relationships are not as strong as those for flux towers (possibly due to large uncertainties in the H estimates), they confirm the positive linkage between ΔH and ΔCloud shown in Fig. 3b.

Fig. R4. The regression between ΔCloud and ΔH from (a) satellite and (b) CLM. The correlation coefficient refers to the spearman correlation coefficient.

In the revision, we performed new analyses on the seasonality of ΔH and ΔCloud and their close match further supports the sensible heat explanation. Please refer to Fig. R12 and the new text below.

Fig. R12 (Figure S11). The seasonal changes of ΔH from flux sites and MODIS ΔCloud for three selected regions: (a) Southeast US (97°W to 75°W, 30°N to 40°N), (b) Europe (10°E to 30°E, 47°N to 55°N), and (c) Amazon (70°W to 50°W, 16°S to 5°S). Months with snow cover are shown as shaded areas in Panels a-c. The n in each panel indicates the number of flux sites within each selected region (Table S1).

Further evidence comes from closely tracked seasonality between ΔH and ΔCloud at paired flux sites (Fig. S11). The larger sensible heat over forest corresponds to greater cloud enhancement (e.g., Europe) whereas the lower sensible heat over forest corresponds to stronger cloud inhibition (e.g., Southeast US and Amazon).

In summary, the linkage between ΔH and ΔCloud is confirmed by three independent datasets in either qualitative or quantitative ways and the collective evidence supports the sensible heat explanation.

(2) We also notice the inconsistency in the cloud effects estimated from different data sources. The estimated cloud effects can differ in magnitude and even sign between different cloud datasets. Since cloud cover datasets from different satellites have their own strengths, limitations, and uncertainties, we used multiple cloud datasets to estimate cloud effects to reduce the data uncertainty. There are indeed differences in the cloud effects in central Africa where MSG data showed strong inhibition effects while MODIS data showed mixed positive and negative cloud effects. Additionally, the cloud effect estimated from Sentinel-5P pointed out cloud inhibition in

central Africa (Fig. S13), though we did not report Sentinel-5P in the main text due to its short time period and different cloud amounts compared to MODIS and MSG data.

To evaluate the cloud cover data, we compare cloud cover fractions of MODIS, MSG, and Sentinel against those from the MODIS cloud property product (MCD06COSP_D3_MODIS) at 1 deg (<https://ladsweb.modaps.eosdis.nasa.gov/missions-and-measurements/science-domain/cloud/>). MCD06COSP_D3_MODIS is selected as the reference dataset because it is widely used in atmosphere and climate science and is deemed to be a reliable source. The scatterplot in Fig. R3 shows that MODIS and MSG cloud cover fractions are closer to the 1:1 line, while Sentinel cloud fractions deviate from the 1:1 line and show clear underestimation biases relative to the reference. This comparison suggests that MODIS and MSG are more reliable sources for cloud cover fraction. Therefore, we did not rely on Sentinel data and only used it for cloud type classification.

Fig. R13. The scatterplot between cloud cover fractions from MODIS/MSG/Sentinel and the referenced MCD06COSP_D3_MODIS data. The solid line represents the regression line for each dataset, while the black line refers to the 1:1 line.

The specific reason for the inconsistency among data sources is currently unknown but we do agree that cloud data present a key source of uncertainty in our results and worth further discussion. In this case, it is important to have independent evidence from other studies for verification. For the Amazon, there is ample observation evidence from other research to corroborate the estimated cloud effects and their seasonality. However, such evidence for central Africa is lacking. Thus, further research is needed in these understudied regions to resolve this issue.

We added a discussion on this matter in the revision:

However, it is worth noting that the estimated cloud effects, despite their broad agreement in the global pattern across datasets, can differ in magnitude and even in sign in certain regions (e.g., the inconsistent cloud effects and their seasonality in central Africa), suggesting cloud data to be a key uncertainty source for our analysis. This is exacerbated by the lack of direct observations in regions like central Africa which have received little attention compared to the Amazon,

hindering comparisons against other available evidence. This highlights the importance of dedicated observational efforts in specific regions, especially those understudied, to provide complementary information to our global-scale analysis.

(3) During the revision, we noted that the global study by Guillod 2015 identified several regions showing temporal preference of afternoon rain with drier soil conditions (Central US, West Amazon, and parts of the Sahel and Equatorial Africa). It is interesting that these regions roughly correspond to cloud inhibition found in our study. We added a paragraph to discuss how the mesoscale mechanism in our study connects to a large body of research in land and atmosphere interaction focusing on surface fluxes and soil moisture anomalies impacts on precipitation.

The mesoscale circulation mechanisms associated with the inhibited clouds over forests echo a large body of research in land-atmosphere interactions emphasizing the critical role of surface fluxes and soil moisture anomalies in the atmospheric boundary layer, clouds, and precipitation processes^{45,46}. The preference of convective clouds and precipitation over drier soils identified in previous studies using observations^{47,48}, analytic models⁴⁹, and numerical simulations⁵⁰ is in line with the enhanced clouds over regions of higher sensible heat flux, as we show here for forest and non-forest transitions. Interestingly, the preference of clouds over non-forest found in Southeast US, Amazon, and Central Africa of our study also roughly correspond to those reported to exhibit a temporal preference of afternoon rain with drier soil conditions (Central US, West Amazon, and parts of the Sahel and Equatorial Africa)⁴⁸.

b) The mechanism proposed by the authors is established in Figure 4. Based on figure 4a, the authors indicate that forests with enhanced cloud cover have both higher H and LE compared to non-forested regions, which then leads to higher ABL and more LCL crossings. I am a bit confused by this since, sensible and latent heat fluxes together tend to balance the net radiation.

What's shown in the schematic Figure 4 is a highly simplified and generalized pattern. Fig. 4a indicates two pathways by which forests enhance cloud cover: the high latent heat (ET) over the forest provides moisture, and the larger sensible heat provides a lifting mechanism for cloud formation. This is supported by our own data and the literature.

In most forests, both sensible heat and latent heat are higher than nonforest (as shown in Figs 3, S10). Moreover, according to results by Duveiller 2018, the conversion of forests to crops/grasses leads to a reduction in both latent heat and sensible heat (plus ground flux), suggesting that forests have larger latent heat and sensible heat than nonforest (Fig. R14). It is true that sensible heat and latent heat fluxes are balanced by net radiation, but forests usually have larger net radiation than nonforest due to their lower albedo (less reflected shortwave radiation) and smaller upward longwave radiation (cooler surface).

Fig. R14 excerpted from Duveiller et al 2018. Global summary of the mean annual potential change in surface energy balance and temperature for various transitions in vegetation type as derived from satellite observations.

However, there are regional and seasonal variations. For example, some tropical forests have overall higher ET and lower H than nonforest, or higher H and lower ET than nonforest during the dry season, and some boreal forests have slightly lower ET and higher H than nonforest.

Our results in Fig. R15 also show that forests with enhanced cloud cover have higher sensible heat and larger ET than nonforest, while forests with inhibited cloud cover have lower sensible and larger ET than nonforest,

Fig. R15. Sensible (H, unit W/m²) and latent heat flux (ET, unit mm/8day) differences between forest and nonforest in JJA for regions with cloud enhancement and cloud inhibition. Here, the data of ΔH is estimated from satellite (the same for Figure 3b) and ET data was the same as Figure S10.

Despite the difference in albedo, I am not sure that I would expect sensible heat fluxes in non-forested environments to be lower than forest H (especially in tropical regions) and forest ABLs to be higher. For example Fisch et al (2004) cited by the authors does not show ABL height differences during the wet season and higher ABL and H over pasture compared to forest during the dry season, which contradicts the authors stated mechanism for Amazonia.

There might be some misunderstanding on this part. As shown by our results in Fig. 3b, for the inhibited cloud over forests in the Amazonia regions during the dry season, sensible heat fluxes in non-forest are actually higher (NOT lower) than forests. This causes the gradient in sensible heat between forest and non-forest (high over nonforest/pasture and low over forests) and it creates a higher ABL over pasture than forests and thus promotes cloud development. This is consistent with

Fisch et al (2004) that “during the dry season. In this period the sensible heat fluxes are very high over pasture, creating a CBL around 550m deeper compared to that over the forest.” This effect is weaker in the wet season because of the smaller differences in sensible heat and ABL between forests and pasture. Therefore, the interpretation of our results are supported by Fisch et al (2004).

Given the fact that cloud cover itself affects the energy supply, I feel that a deeper dive into the data is needed to 'prove this relationship'. For boreal regions, it may also not be true, since I would expect wetlands to have a similar (or even lower albedo) compared to forests. The authors should use the data to also look into ET/LE which is not presented here.

I feel that the mechanisms shown in figure 4b is fairly well established in the literature.

(1) In the revision, we added new evidence from seasonal variations between ΔH and ΔCloud which lends further support to the sensible heat mechanism (Fig. R12; please see more details in our response to previous comments).

(2) In the initial manuscript, we had included a figure for the ET differences between forest and nonforest (ΔET) in the supplementary information which is attached below. Fig. R16 shows most forests have higher ET than nonforest and such an ET surplus over forest declines with latitude. In boreal regions, the ET difference is small and can be slightly negative. These patterns are consistent with other studies (Duveiller et al 2018; Li et al 2015). However, as noted in the manuscript, the differences in ET are unable to well explain the cloud effects, due to its mismatched spatial patterns with ΔCloud . Moreover, at the paired flux sites, ΔCloud has a much stronger relationship with ΔH ($r > 0.3$) than with ΔLE ($r \approx -0.1$).

Fig. R16 (Figure S10). The differences between forests and non-forest in LST (a), ET (c), and soil moisture (e) in JJA from 2002 to 2018 and their latitudinal patterns (b,d,f).

(3) In particular, wetland was excluded in the calculation of differences between forest and nonforest in cloud cover ($\Delta Cloud$) and other biophysical variables (ΔET and ΔSM) based on MODIS land cover data, as described in methodology. The confounding effect of wetland on albedo, cloud cover, and other variables can be minimized.

(4) It is true that the mechanisms summarized in Fig. 4 are well established in the literature especially for the Amazon regions. The innovation of our study is that we expand the limited regions in earlier studies to the global scope and we found that the inhibited cloud cover previously observed in the Amazon can also be found elsewhere in the world, and such spatial pattern is driven by the sensible heat differences. Our data-driven assessment allows for a more

comprehensive understanding of the effect of forests on clouds at a global scale and improves the knowledge on forest-cloud interactions.

c) The differing behavior of EurAsia may be due to differences in the land-cover pairs. Based on Figure S9 a large portion of the nonforested land cover in the boreal forests is wetlands, which have a very different surface energy balance compared to grasslands and croplands, which are found in the other regions. It may thus make sense to formulate 2 mechanisms rather than trying to combine everything into a single theory. It may make sense to limit the scope of the paper to tropical regions (especially since there seems to be a clear signal in the Amazon irregardless of the cloud cover sensor).

It is important to clarify that by using MODIS land cover data, wetland pixels were excluded when calculating the differences between forest and nonforest in cloud cover (ΔCloud) and other biophysical variables (ΔET and ΔSM) with the window searching method. This was noted in the method section as:

For unchanged non-forest, pixels classified as water, snow/ice, or wetland were excluded using the major composite of MODIS land cover from 2002 to 2005 with the International Geosphere-Biosphere Programme (IGBP) classification scheme.

Therefore, the confounding effect of wetland on ΔCloud is minimized.

However, the presence of wetland for dominant non-forest land cover type shown in Fig. S9 was due to the use of ESA land cover data. Although "wetland" pixels were excluded based on MODIS land cover data, there are some non-wetland pixels classified as "wetland" by the ESA land cover data, due to the differences in land characterization between the two. In theory, there could be two versions of dominant land cover types, one for MODIS (not shown) and another for ESA land cover data (as shown in Fig. S9). The version by MODIS will not include wetland. We chose to present the ESA land dataset because it was used in the calculation of ΔH which needs matching specific forest and nonforest land cover types. Moreover, the ESA land cover data were used to produce the satellite estimate of $\text{H}+\text{G}$. Nevertheless, the ESA land cover data and their dominant land cover types for estimating ΔH do not affect the interpretation of cloud effects. First, as we noted in the manuscript, "The geographic variations in specific land cover types of the global forest and non-forest vegetation show little spatial resemblance to \$\Delta\text{Cloud}\$ (Fig. S9)." For the boreal regions where wetland was shown to be the dominant land cover for nonforest in Fig. S9, their cloud effects did not show spatial discontinuity related to wetland. This implies that specific land-cover pairs are not the main factor driving the spatial variations of the cloud effects. Second, wetland was not used in deriving ΔH from CLM because there is no wetland land unit in CLM. The results based on ΔH from CLM also support a similar spatial correspondence between ΔH and cloud inhibition as those with ΔH from satellite estimates.

d) I am wondering to what extent cloud cover as derived by Modis is a good category for analysis, since it is a simple yes/no mask with no additional information related to type of cloud, cloud water, fractional cover, etc. The Sentinel data seems to provide a much richer set of information, but also complicates the analysis tremendously.

Thanks for this comment.

(1) Sentinel data supply richer information than cloud cover fraction, but as you said, we are not planning to dive deep into it as it will complicate our analysis. We mainly used Sentinel data for cloud type classification.

(2) We do agree that the quality of cloud cover data is an important issue for our study. Since different cloud cover datasets have their own strengths, limitations, and uncertainties, we used multiple cloud datasets to estimate cloud effects to reduce the data uncertainty. To evaluate the cloud cover data, we compare cloud cover fractions of MODIS (binary cloud mask), MSG, and Sentinel against those the MODIS cloud property product (MCD06COSP_D3_MODIS, at 1 deg) (<https://ladsweb.modaps.eosdis.nasa.gov/missions-and-measurements/science-domain/cloud/>). MCD06COSP_D3_MODIS is selected as the reference dataset because it is widely used in atmosphere and climate science and is deemed to be a reliable source. The scatterplot in Fig. R13 shows that MODIS (binary cloud mask) and MSG cloud cover fractions are closer to the 1:1 line, while Sentinel cloud fractions deviate from the 1:1 line and show clear underestimation biases relative to the reference. This comparison suggests that MODIS and MSG are more reliable sources for cloud cover fraction. Therefore, we did not rely on Sentinel data for quantifying cloud effects due to its short time period and underestimation compared to MODIS and MSG data, it is only used for cloud type classification.

Fig. R13. The scatterplot between cloud cover fractions from MODIS/MSG/Sentinel and the referenced MCD06COSP_D3_MODIS data. The solid line represents the regression line for each dataset, while the black line refers to the 1:1 line.

2) Organization of the manuscript

The attribution section feels out of place. I would consider this data preparation and would move this to the methods section and then only use the data without potential orthographic effects.

Thanks for the suggestion.

(1) In the revision, to make the story more coherent, the attribution section is no longer a separate section. The original Figure 2 and the corresponding text about cloud effect attribution were merged to the section "Potential effects of forests on cloud cover".

Please check the revised text and the updated Figure 1 below:

The estimated cloud effects of forests could be confounded by orographic clouds because of the dual influences of topography on forest distribution and cloud formation. Human impacts resulted

in a global tendency of existing forests to be located at more complex terrain with a higher elevation than non-forest²⁹. The high elevation per se could facilitate cloud formation through the orographic lifting of moist air³⁰, leading to increased cloud cover over the forest located at a higher altitude (Fig. S5). To isolate the orographic cloud effect, we decompose ΔCloud into contributions of tree cover and elevation (see methods, Fig. S6), and we find that the global pattern of ΔCloud is dominated by tree cover induced cloud effects (41% grid boxes for cloud enhancement and 22% for cloud inhibition), followed by elevation induced cloud effects (30%), and unexplained effects due to other factors (7%) (Fig. 1f). This confirms that most of the observed cloud effects are robust features attributable to tree cover rather than topography and other factors.

Fig. R11 (Figure 1). The potential effects of forests on June-August (JJA) cloud cover fraction and their attribution. The potential effect is defined as the differences in cloud cover fraction between forests and nearby non-forest (ΔCloud) from MODIS and MSG satellites that detect clouds. (a) Potential effects of forests on cloud cover fraction based on MODIS data from 2002 to 2018 (overpass at 13:30 local time) and (b) their latitudinal patterns with cloud enhancement and

inhibition effects separated. (c,d) Potential effects of forests on cloud cover fraction based on hourly MSG data from 2004 to 2013 (overpass at 14:00 local time) and (e) the timing of the maximum effect during a day. The numbers in Panels (b) and (d) show the percentage of cloud enhancement (red) and inhibition (blue). (f) Attribution of cloud effects of forests to tree cover and elevation based on MODIS and MSG data. The five attribution categories include tree cover induced cloud increase (Tree+) and decrease (Tree-), orography induced cloud increase (Orography+) and decrease (Orography-), and other unexplained effects. The percentage of each attribution category is calculated based on the MODIS results.

(2) Removing the confounding orographic effects is an important measure to ensure reliable estimates of the cloud effect. For this purpose, in the data preparation section, we masked out forest and nonforest comparison samples from regions with complex terrain. The reviewer suggests to mask out cloud effects attributable to topography in the presentation. Here we created a figure below for demonstration (Fig. R17). It can be seen that the spatial pattern of ΔCloud after masking is basically the same as the original figure except for slightly fewer grids. Although this masking can be done for Figure 1a, implementing it throughout the paper will create additional problems. This is because the attribution analysis shown in the original Fig. 2 was performed for the cloud effects in JJA. The attribution results in JJA may not be the same for other months owing to diurnal and seasonal variations in the cloud effect. If implementing the masking throughout the manuscript, we would need to repeat the attribution for each month, and even every hour for MSG data to remove those orography grids. This makes the processing too complicated and actually gives very few benefits. Besides, this practice also affects the sensible heat differences part because they need to match the grid boxes of cloud effects. For these considerations, we decided not to implement this masking method.

Fig. R17 Cloud effects of forests after masking out the orography dominant grids.

3) Presentation of the manuscript

a) The manuscript relies heavily on maps with many small pixels, that show substantial small scale variation (e.g. Figure 1a). It is very difficult to extract quantitative and even qualitative information from these maps. I would encourage the authors to think, how they can synthesize this information into better formats along the lines of Figure 1b, 3b, Figure S12 etc.

We appreciate these comments about the presentation of the results.

(1) The map figure of Δ Cloud indeed shows small scale variations, which might make it difficult for readers to extract quantitative information. We could not get rid of the small scale variation because that is what the data show. The difficulty might partially arise from the use of a continuous colormap which makes extracting specific values from the colors difficult. In the revision, we replaced the continuous colormaps by discrete colormaps in all main figures. With this change, a given color represents a specific value range of a variable, and it makes it easier to extract quantitative information. Moreover, we also merged the attribution figure to Figure 1. The four colors of the attribution figure represent a combination of influencing factors and signs of cloud effect, therefore, it can effectively smooth out small variations and help interpret the spatial pattern of Δ Cloud.

(2) We also redesigned the seasonal variations figure for Δ Cloud and put it as the new Figure 2 in the main text. It provides quantitative information regarding seasonality.

(3) We will also share all the data generated in this work to allow anyone who has an interest to explore more and extract any quantitative information they need.

We hope these changes will help the reviewers and the readers perceive the results.

b) The authors should ensure that abbreviations are introduced in the text (and not only in the methods).

We have checked and defined the abbreviations.

References

Chen, L., & Dirmeyer, P. A. (2020). Reconciling the disagreement between observed and simulated temperature responses to deforestation. *Nature Communications*, 11(1), 202. <https://doi.org/10.1038/s41467-019-14017-0>

Degano M F, Rivas R E, Carmona F, et al. Evaluation of the MOD16A2 evapotranspiration product in an agricultural area of Argentina, the Pampas region[J]. *The Egyptian Journal of Remote Sensing and Space Science*, 2021, 24(2): 319-328.

Duveiller, G., Hooker, J., & Cescatti, A. (2018). The mark of vegetation change on Earth's surface energy balance. *Nature Communications*, 9(2018). <https://doi.org/10.1038/s41467-017-02810>

Faisal A, Novita E. An evaluation of MODIS global evapotranspiration product (MOD16A2) as terrestrial evapotranspiration in East Java-Indonesia[C]//IOP Conference Series: Earth and Environmental Science. IOP Publishing, 2020, 485(1): 012002.

Guillod, B. P., Orlowsky, B., Miralles, D. G., Teuling, A. J., & Seneviratne, S. I. (2015). Reconciling spatial and temporal soil moisture effects on afternoon rainfall. *Nature Communications*, 6(1), 6443. <https://doi.org/10.1038/ncomms7443>

Kim H W, Hwang K, Mu Q, et al. Validation of MODIS 16 global terrestrial evapotranspiration products in various climates and land cover types in Asia[J]. *KSCE Journal of Civil Engineering*, 2012, 16(2): 229-238.

Kleidon, A., Fraedrich, K., & Heimann, M. (2000). A green planet versus a desert world: Estimating the maximum effect of vegetation on the land surface climate. *Climatic Change*, 44(4), 471–493. [https://doi.org/Doi 10.1023/A:1005559518889](https://doi.org/Doi%2010.1023/A:1005559518889)

Li, Y., Zhao, M., Motesharrei, S., Mu, Q., Kalnay, E., & Li, S. (2015). Local cooling and warming effects of forests based on satellite observations. *Nature Communications*, 6, 6603. <https://doi.org/10.1038/ncomms7603>

Riemann-Campe, K., Fraedrich, K., & Lunkeit, F. (2009). Global climatology of Convective Available Potential Energy (CAPE) and Convective Inhibition (CIN) in ERA-40 reanalysis. *Atmospheric Research*, 93(1–3), 534–545. <https://doi.org/10.1016/J.ATMOSRES.2008.09.037>

Ruhoff A L, Paz A R, Aragao L, et al. Assessment of the MODIS global evapotranspiration algorithm using eddy covariance measurements and hydrological modelling in the Rio Grande basin[J]. *Hydrological Sciences Journal*, 2013, 58(8): 1658-1676.

Teuling, A. J., Taylor, C. M., Meirink, J. F., Melsen, L. A., Miralles, D. G., van Heerwaarden, C. C., ... de Arellano, J. V.-G. (2017). Observational evidence for cloud cover enhancement over western European forests. *Nature Communications*, 8, 14065. <https://doi.org/10.1038/ncomms14065>

Winckler, J., Lejeune, Q., Reick, C. H., & Pongratz, J. (2019). Nonlocal Effects Dominate the Global Mean Surface Temperature Response to the Biogeophysical Effects of Deforestation. *Geophysical Research Letters*, 46(2), 745–755. <https://doi.org/10.1029/2018GL080211>

REVIEWER COMMENTS

Reviewer #1 (Remarks to the Author):

In their manuscript "Contrasting impacts of forests on cloud cover based on satellite observations", the authors present an analysis of satellite cloud cover, where they compare the frequency of cloud formation over closely co-located forested and non-forested regions. I commend the authors on their revision of the manuscript, as many of the sections which I found puzzling during the first round of review are now much clearer, and changes made to the figures have also improved their clarity.

I strongly suggest the authors include a comparison of their results to a very similar study recently published in Nature Communications (Duveiller et al. 2021), to clearly lay out the similarities/differences in their methodological approach which may contribute to the similarities/differences in their conclusions. Specifically, at the abstract/main conclusions level, this study (the one being reviewed now) finds that tropical forests on the whole reduce cloud cover while the Duveiller study seems to find that forests increase cloud cover in most tropical regions (though both studies seem to find a cloud reduction with increased forest in the south Amazon – so this may be more of a "what the authors choose to focus on / not terribly specific with wording" problem vs. an actual difference in raw results – this is why an explicit comparison with this very similar paper would be useful!)

I have a few minor comments on the revised manuscript, which I note below.

Line 34: "We find enhanced clouds over most temperate and boreal forests but inhibited clouds over Amazon, Central Africa, and Southeast US" – how does this compare to the work of, e.g. Bonan 2008, which finds that forests support tropical cloud cover?

Figure 2: this is minor, and the authors can ignore it if they choose, but I was confused about where panels b/c/d/e were representing, just because the location labels weren't all in the same place. It would be easier (for me) if the labels were all in the same location on each subplot, or part of the title.

Line 145 / Figure 2 caption: "Months with snow cover are shown as the shaded regions" – is this correct? E.g. Every single month on the blue line of d – the Amazon – has shading, but ... the Amazon isn't snowy year round. Or really, ever... OH! You mean the gray shading! Maybe add "gray shading" to the caption, then also state what the red/blue shading is! (Some kind of statistical spread, I assume)

Line 241: I think this is a really important point!

Line 569: I was unable to access data at this link / searching figshare for this doi. Perhaps the authors just haven't made the data publicly available yet, but as such I cannot provide feedback on if the data meets with FAIR and Nature Communication's data availability requirements.

Reviewer #2 (Remarks to the Author):

I would like to thank the authors for taking the time to revise the paper and to respond to my comments. I appreciate the addition of the new figure 2, which in conjunction with Figure 5 help visualize the potential effects of forest cover on clouds.

One of my earlier criticisms was that much of the information in the paper is conveyed through comparison of small maps, and the new figure 2 as well as figure 5 are two examples, where other types of figures are used effectively.

I am still wondering about figure 4, which summarizes the proposed mechanism and I am wondering whether this is the best use of this figure (especially given the figure limitations in the journal). I feel that the role of mesoscale circulations in 4b is well understood. The discussion of 4a also mentions mesoscale circulations, which I are are labeled 'forest breeze' in 4a. However, I am not sure how important these would be as opposed to the much higher LE from the forest compared to surrounding areas. This is then further complicated by the fact that cloud cover reduces surface fluxes. It may be more effective to use this space to bring in an additional figure that is based on data.

Lastly, I am still concerned about the fact that the reported effects are very small (especially when considering that positive and negative enhancements of similar magnitude are present at all latitudes (proxy for climate), Figure R 5) and that there is disagreement in the sign of the effect for Southeast US and Central Africa (Figure S13). I am therefore wondering, whether there is any method that could be applied to help understand to what extent the results found are robust.

We are grateful for the additional comments from the two reviewers. We have further revised the manuscript and provided a point-to-point response to each comment. The original comments are in **bold** font, our response is in regular font, and the changes in the text are in blue.

Reviewer #1 (Remarks to the Author):

In their manuscript “Contrasting impacts of forests on cloud cover based on satellite observations”, the authors present an analysis of satellite cloud cover, where they compare the frequency of cloud formation over closely co-located forested and non-forested regions. I commend the authors on their revision of the manuscript, as many of the sections which I found puzzling during the first round of review are now much clearer, and changes made to the figures have also improved their clarity.

We thank the reviewer for commending our efforts for revision. We appreciate the constructive comments provided by the reviewer, which improved the work significantly.

I strongly suggest the authors include a comparison of their results to a very similar study recently published in Nature Communications (Duveiller et al. 2021), to clearly lay out the similarities/differences in their methodological approach which may contribute to the similarities/differences in their conclusions. Specifically, at the abstract/main conclusions level, this study (the one being reviewed now) finds that tropical forests on the whole reduce cloud cover while the Duveiller study seems to find that forests increase cloud cover in most tropical regions (though both studies seem to find a cloud reduction with increased forest in the south Amazon – so this may be more of a “what the authors choose to focus on / not terribly specific with wording” problem vs. an actual difference in raw results – this is why an explicit comparison with this very similar paper would be useful!)

Thanks for this suggestion. We also noticed the work of Duveiller 2021 (hereafter D2021 for short) published in Nature Communications during our revision stage. The D2021 paper focused on a similar topic as ours: the effect of forest on cloud cover (afforestation in their context). This reaffirms the critical knowledge gap identified by both and, therefore the value of our study. Similar to ours, D2021 also used satellite-derived cloud cover data to quantify the cloud effects of forest by a space-for-time approach. Specifically, in terms of cloud data, D2021 used a cloud property dataset produced by a third-party research team (Stengel 2017) based on the MODIS reflectance data using their algorithm. In contrast, we used the cloud mask data produced by the official MODIS team, which has been widely used to develop a variety of downstream MODIS products and had undergone rigorous test and validation. In terms of methodology, the cloud effects of D2021 were quantified by using a regression approach adapted from their previous work for land surface temperature and surface energy fluxes (Duveiller 2018). This was done by estimating the slopes/sensitivities of cloud cover to areal fractions of different land cover types and then assuming 100% land conversion between any two land cover types. This method is mathematically

more complicated than our method, and it involves non-trivial effort in post-processing, screening, and filtering the data and results. In contrast, our method, originally proposed to estimate the temperature effect of forests in our previous work (Li 2015), relies on direct comparisons of cloud cover between forest and nearby non-forest and is more intuitive and straightforward.

As for the results, D2021 produced similar cloud effects with a general consistent global pattern with ours. Both studies found that cloud enhancement is the dominant effect of forest/afforestation. However, there are notable regional differences between D2021 and ours, as the reviewer correctly pointed out in the tropical regions.

It is worth investigating what caused these differences. As outlined above, we suspect that cloud data and methodology are the most likely reasons, especially the former. We showed in our work that cloud data are the most important source of uncertainty to the cloud effects.

(1) We first examined the cloud cover data of D2021 and compared with our MODIS data (with different algorithms and processing). We found significant differences between the cloud cover datasets that may lead to different cloud effect estimations. Fig R1 shows the multi-year averaged cloud cover fractions of these two datasets. The global pattern looked similar. A closer look after zooming in (Fig R2), there are visible stitches/scanlines and strips-like patterns over some parts of the ocean and land in D2021 cloud data, which are probably artifacts of satellite images processing. In comparison, the cloud data we used did not have such artifacts. We also observed frequent spatial discontinuities on D2021 data, probably due to water body. For example, the cloud suppression around the Amazon river (due to water body effect) was more evident on our cloud data, while there were some random pixels with high cloud fractions near the river with D2021 data.

(a)

(b)

Fig R1. The multi-year averaged cloud cover fractions between the D2021 (a) and our data (b)

(a) Strip-like pattern in D2021 cloud cover fractions

(b) D2021 in South America

(c) D2021 in North America

(d) Our cloud data in South America

(e) Our cloud data in North America

Fig R2. Zoomed view of the multi-year averaged cloud cover fractions between the D2021 and our data. (a) Strip like pattern in D2021 cloud cover fractions. (b-e) the second row is D2021 data in South America and North America; the third row is our cloud data.

Fig R3. The scatterplot between cloud cover fractions from Duveiller, our MODIS data against the referenced MCD06COSP_D3_MODIS data (one degree). The solid line represents the regression line for each dataset, while the black line refers to the 1:1 line.

Despite the above visual inspection, we also compared cloud cover fraction of D2021 against MCD06 in Fig R3 (MCD06 was used for validation in the previous response letter because it is recognized as an accurate source of cloud cover). According to the regression line, the D2021 cloud fractions deviated from the 1:1 line of MCD06 more than our cloud data. Besides, the D2021 cloud fractions were much more scattered. This revealed a large discrepancy in cloud fractions of D2021 against the reference data, which might be caused by the spatial discontinuity we observed on the map.

(2) To explore the influence of methodology, we applied our method to the D2021 cloud data to estimate the cloud effects in JJA (Fig R4a) and compared them with results of D2021 (D2021 data and their method, reproduced with our color scheme in FigR4(b)). Despite some differences in spatial coverage, the spatial patterns produced by different methods were very similar. The greater magnitude of FigR4b was due to the assumption of 100% land conversion (100% forest pixels with 100% non-forest types like grass, crops.) in the D2021 method. This comparison suggests that the method was not the key factor to explain the different cloud effects.

(a) Cloud effects in JJA estimated using the D2021 data with our method

(b) Cloud effects in JJA estimated by D2021 (D2021 data and D2021 method)

Fig R4 (c) Cloud effects in JJA estimated by our data and our method

To explore the influence of cloud data, we compared cloud effects estimated using the D2021 data with our method (Fig R4a) and our results (our data and our method, FigR4c). Although the global pattern was similar, the different cloud data led to some discernable differences at regional scales. Compared with our data, D2021 data produced more cloud enhancement with a more uniform pattern in Europe. The cloud

inhibition we identified in Southeast US was still visible with the D2021 data, but its coverage was reduced. In the tropics, the cloud inhibition in the Amazon was similar, but cloud inhibition did not appear in central Africa of D2021 data. However, the cloud inhibition in central Africa was evident with the MSG cloud data reported in our manuscript.

(3) Although there is inconsistency in cloud effects estimated across cloud datasets, the regional cloud inhibition found in our study is supported indirectly by the mesoscale circulation mechanism as well as the actual cloud impact of forest loss. On the one hand, the occurrence of cloud inhibition (e.g., Amazon, Southeast US, and Central Africa) corresponded to locations with lower sensible heat over forests based on two independent sensible heat data, which provide favorable conditions for the development of mesoscale circulation and inhibited cloud over forests. On the other hand, the cloud inhibition in Southeast US suggests that the forest loss in this area would lead to cloud cover increase. Consistent with this inference, we found that the actual impact of forest loss hotspot in Southeast US indeed increased cloud cover. This consistency between the potential and actual impacts of forests indirectly supports the robustness of cloud inhibition in the Southeast US. However, not all cloud inhibition regions have forest loss hotspots like Southeast US.

To conclude, the above analyses demonstrate that cloud data were the primary cause of the different cloud effects between D2021 and ours. This again reinforces our argument in the manuscript that “cloud data are a key uncertainty source for our analysis” since in our results we observed regional differences in the cloud effects estimated from MODIS, MSG, and sentinel cloud data. Given this uncertainty from cloud data, the cloud effects estimated relying on a single source would suffer from this uncertainty. Therefore, it is imperative to use multiple different cloud data to reduce such uncertainty and extract the robust signals, as we did for our analyses. This is also the strength of our work compared to other work. Moreover, we offered mechanistic explanations by the sensible heat differences that support the observed spatial pattern of cloud effects. For example, the negative ΔH found in Amazon, central Africa, southeast US from independent sources was in line with the cloud inhibition in these regions (albeit the strength of the signals varies across different cloud data sources). The consistency between cloud inhibition and actual cloud increase (e.g., in Southeast US) following forest loss further improve estimation credibility. The direct evidence from different cloud data and the indirect evidence from mechanism explanation and the actual impact of forest loss jointly improve the credibility of our results. As we emphasized, more fine-scale observations are needed to go beyond the global pattern shown, and dive into regional and specific cloud impact. In this way, more studies like D2021 are welcome additions to advance our understanding of forest and cloud interactions.

In the revision, the comparison with D2021 paper and cloud effect uncertainty was discussed:

It is worth noting that the estimated cloud effects, despite their broad agreement in the global pattern across datasets, can differ in magnitude and even in sign in certain regions (e.g., the inconsistent cloud effects and their seasonality in central Africa), suggesting cloud data are a key uncertainty source for our analysis. The cloud effects of forests estimated in a recent study⁵⁷ based on a different MODIS-derived cloud dataset⁵⁸ (produced by a different retrieval algorithm) revealed a similar global dominance of cloud enhancement and regional prevalence of cloud inhibition (e.g., in Amazon). Yet regional inconsistencies remained for Southeast US and Central Africa, where the identified cloud inhibition could be weakened or absent with alternative cloud data. This emphasizes the need of using multi-source cloud data to improve the robustness of the estimated cloud effects while reducing the uncertainty from data. Nevertheless, the occurrence of cloud inhibition in these regions is indirectly supported by lower sensible heat over forests, which is in line with the mesoscale circulation mechanism, as well as the consistency between potential cloud inhibition and the actual cloud increase over forest loss locations (i.e., Southeast US hotspot). While more direct observational evidence is always desirable to help resolve inconsistencies, the lack of observations in regions like Central Africa, which have received less attention than the Amazon, hinders comparisons against other available evidence. This highlights the importance of dedicated observational efforts in specific regions, especially those understudied, to provide complementary information to our global-scale analysis.

I have a few minor comments on the revised manuscript, which I note below.

Line 34: “We find enhanced clouds over most temperate and boreal forests but inhibited clouds over Amazon, Central Africa, and Southeast US” – how does this compare to the work of, e.g. Bonan 2008, which finds that forests support tropical cloud cover?

We are not sure which Bonan 2008 work was mentioned by the reviewer. The following response assumes it was the review paper by Bonan published in *Science* in 2008 cited in our manuscript (Forests and climate change: Forcings, feedbacks, and the climate benefits of forests. *Science*. 320, 1444–1449 (2008)). This paper reviewed the progress on the role of forests on climate system by the time it was published. The review by Bonan (2008) suggested that the impact of tropical forests on clouds is scale-dependent. At large spatial scales, tropical forests support precipitation and cloud cover, as indicated by Fig 3 of Bonan (2008). However, at small spatial scales, the biophysical impacts of forests on clouds become different and more complex compared to large scales, as he wrote: “*However, forest-atmosphere interactions are complex, and small-scale, heterogeneous deforestation may produce mesoscale circulations that enhance clouds and precipitation.*”

The objective of our work is to use global satellite observations to study the forest-cloud interactions at small-scale and provide new insight. Our understanding has progressed significantly since Bonan (2008) by more recent regional studies such as Teuling (2017) and Wang (2009). Our study is in line with these regional studies

showing cloud enhancement in temperate regions (Teuling 2017) and others showing small-scale cloud inhibition for the Amazon (Wang 2009).

Figure 2: this is minor, and the authors can ignore it if they choose, but I was confused about where panels b/c/d/e were representing, just because the location labels weren't all in the same place. It would be easier (for me) if the labels were all in the same location on each subplot, or part of the title.

Thanks for this suggestion. In the revised figure, we slightly changed the design and placed the location labels to the same lower left corner of each subplot.

Fig R5 (Fig 2). Seasonal variations of the potential effects of forests on cloud cover. Months with the maximum magnitude of ΔCloud for MODIS and MSG during the snow-free season are shown on the global map. Seasonal variations of ΔCloud are shown in selected regions: (a) Southeast US (97°W to 75°W, 30°N to 40°N), (b) Europe (10°E to 30°E, 47°N to 55°N), (c) Amazon (70°W to 50°W, 16°S to 5°S), and (d) Central Africa (10°E to 33°E, 7.5°S to 0°) for MODIS and MSG. Months with snow cover are shown as the grey shaded areas in the background. The blue and red shaded areas denote the 95% confidence interval of ΔCloud for MODIS and MSG.

Line 145 / Figure 2 caption: “Months with snow cover are shown as the shaded regions” – is this correct? E.g. Every single month on the blue line of d – the Amazon – has shading, but ... the Amazon isn’t snowy year round. Or really, ever... OH! You mean the gray shading! Maybe add “gray shading” to the caption, then also state what the red/blue shading is! (Some kind of statistical spread, I assume)

Sorry for the confusion. We clarified this in the figure caption as:

Months with snow cover are shown as the grey shaded areas in the background. The blue and red shaded areas denote the 95% confidence interval of Δ Cloud for MODIS and MSG.

Line 241: I think this is a really important point!

Thanks, we also think this is an interesting and potentially important finding.

Line 569: I was unable to access data at this link / searching figshare for this doi. Perhaps the authors just haven’t made the data publicly available yet, but as such I cannot provide feedback on if the data meets with FAIR and Nature Communication’s data availability requirements.

The figshare DOI was reserved for data sharing after publication. As requested by the reviewer, we have made the data link accessible since now (doi:10.6084/m9.figshare.15081510).

Reviewer #2 (Remarks to the Author):

I would like to thank the authors for taking the time to revise the paper and to respond to my comments. I appreciate the addition of the new figure 2, which in conjunction with Figure 5 help visualize the potential effects of forest cover on clouds.

We are grateful for the constructive comments provided by the reviewer, which significantly improved the work.

One of my earlier criticisms was that much of the information in the paper is conveyed through comparison of small maps, and the new figure 2 as well as figure 5 are two examples, where other types of figures are used effectively.

I am still wondering about figure 4, which summarizes the proposed mechanism and I am wondering whether this is the best use of this figure (especially given the figure limitations in the journal). I feel that the role of mesoscale circulations in 4b is well understood. The discussion of 4a also mentions mesoscale circulations, which I are are labeled 'forest breeze' in 4a. However, I am not sure how important these would be as opposed to the much higher LE from the forest compared to surrounding areas. This is then further complicated by the fact that cloud cover reduces surface fluxes. It may be more effective to use this space to bring in an additional figure that is based on data.

Thanks for the comment. We would like to explain our motivation for designing this

figure. We agree that there have been plenty of studies on mesoscale circulation and the mechanisms are relatively well understood. The mechanistic explanation of mesoscale circulation might be familiar to readers in the area of land-atmosphere interaction and boundary layer meteorology. For the broader readership of this journal, we feel this explanation is still necessary and helpful for readers from other related fields like forestry, hydrology etc to better understand these processes. Our previous experience with such mechanical explanation is that having a schematic figure is a much more driven effective way of conveying information than text. Similar schematic figures have been frequently seen in relevant papers (some examples in Fig R6).

Figure 5 | Transition in the dominant convective regime with increasing scales of deforestation. a. In the early period, convection over the

Figure 3 Effects of tropical deforestation on rainfall. Deforestation leads to reduced evapotranspiration (E), resulting in a warmer land surface

Figure 4 | Possible pathways of temperate forest impact on convective cloud formation. Background shading indicates potential temperate

Figure 5 | Atmospheric dynamics induced by the boundary between forest and non-forest. Rainfall may be caused by convection that develops at the

Teuling 2017

Lawrence 2014

Fig R6. Some schematic figures in the relevant literature

As for the mechanisms of cloud enhancement, “forest breeze” was added following the findings of Teuling (2017). They found evidence of a forest-breeze in the specific patterns of cloud occurrence in Landes forest, which is driven by differences in sensible heat fluxes. In fact, the processes driving enhanced clouds over forests are operating at different scales. The forest breeze is mainly at small-scale, while the latent heat and moisture recycling are more important at a larger scale. In the revision, we clarified these mechanisms as well as their operating scales:

The mechanisms of enhanced cloud over forests are associated with several interconnected processes operating at different scales that are conducive to the growth of moist convection (Fig. 4a). Compared with non-forest vegetation, forests usually exhibit high evapotranspiration⁵, which provides abundant water vapor supply

for cloud formation and sustains moisture recycling over large scales^{38,39}. The low albedo and high roughness of forests promote a greater fraction of incoming solar energy partitioned into turbulent heat fluxes, increasing turbulent mixing and convective instability in the boundary layer^{16,29,40}. At small scales, the differential roughness between forest and non-forest induces frictional convergence in downwind direction^{22,41}. Enhanced sensible heating, which typically occurs over the forest relative to non-forest vegetation³⁴, serves as a major lifting mechanism to initiate convection and the growth of boundary layer^{29,32}, and formation of a “forest-breeze” analogous to a sea-breeze. Specifically, the high sensible heat elevates the atmospheric boundary layer (ABL) such that the lifting condensation level (LCL) is lower than the ABL depth^{18,29,39}, thereby supporting low-level cloud formation.

It is true that cloud cover also affects surface fluxes. The enhanced clouds could reduce sensible heat flux over forests, while inhibited clouds could strengthen sensible heat. The cloud effects, once developed, result in dampened flux differences between forest and non-forest, acting as negative feedback. This also involves a memory effect, because cloud differences detected at the satellite overpass time reflect the flux differences accumulated earlier during the morning and noon. The developed cloud effects dampen the flux differences in the afternoon.

We added a discussion on this issue:

Moreover, the cloud effects, once developed, tend to dampen the sensible heat flux differences from which they originate. The inhibited cloud cover (with lower sensible heat) over forests, in turn, enhances sensible heat as the land surface receives more incoming shortwave radiation. It also implies a memory effect as the cloud effects detected at the satellite overpass time reflect the flux differences accumulated earlier during the morning and noon, while the developed cloud effects dampen the flux differences in the afternoon.

Lastly, I am still concerned about the fact that the reported effects are very small (especially when considering that positive and negative enhancements of similar magnitude are present at all latitudes (proxy for climate), Figure R5) and that there is disagreement in the sign of the effect for Southeast US and Central Africa (Figure S13). I am therefore wondering, whether there is any method that could be applied to help understand to what extent the results found are robust. The seemingly “small” cloud effect is related to the range of cloud fraction (from 0~1). The magnitude of the latitudinal averaged cloud effects in Fig 1 varies from ~0.03 to 0. Fig R5 in our previous response letter was only for demonstrative purposes. It showed that the positive and negative effects canceled each other and resulted in fluctuated latitudinal patterns ranging from 0.02 to 0. Moreover, the recent paper by Duveiller 2021 (also mentioned by the 1st reviewer) also estimated the cloud effect of forests which showed a similar magnitude as ours (cloud effect <0.06). For example, the averaged cloud effects by climate zones shown in their Figure 3 (shown below) ranged from 0~0.03 for JJA months (depending on methodology: space-for-time or real forest

cover changes). Since they assumed 100% land conversion in their method, their cloud effects should be greater than ours.

Fig. 3 Comparison between different methods to extract the change in cloud fractional cover (CFrC) following afforestation from satellite records. The seasonality of the Southern Hemisphere was shifted by 6 months to align it with that on the northern one. The error bars represent the standard error

The reviewer’s concern regarding the regional inconsistency in cloud effects across different datasets is legitimate, and we also feel it is a vital issue to address. As shown in our response to 1st reviewer, we carried out additional analyses comparing Duveiller 2021 with our results. It is found that different cloud data lead to different cloud effects at regional scale, especially for cloud inhibition. This again reinforces our argument in the manuscript that “cloud data are a key uncertainty source for our analysis” since we observed regional differences in the cloud effects estimated from MODIS, MSG, and sentinel cloud data. Given this uncertainty from cloud data, the cloud effects estimated relying on a single source would suffer from this uncertainty. Therefore, it is imperative to use multiple cloud data to reduce such uncertainty and extract the robust signals, as we did for our analyses. This is also the strength of our work compared to other work. We also provide indirect supporting evidence to help extract the robust signals from uncertainty of cloud data. The regional cloud inhibition found in our study is supported indirectly by the mesoscale circulation mechanism as well as the actual cloud impact of forest loss. On the one hand, the occurrence of cloud inhibition (e.g., Amazon, Southeast US, and Central Africa) corresponded to locations with lower sensible heat over forests based on two independent sensible heat data, which provide favorable conditions for the development of mesoscale circulation and inhibited cloud over forests. On the other hand, the cloud inhibition in Southeast US suggests that the forest loss in this area would lead to cloud cover increase. Consistent with this inference, we found that the actual impact of forest loss in Southeast US hotspot indeed increased cloud cover. This consistency between the potential and actual

impacts of forests indirectly supports the robustness of cloud inhibition in Southeast US. However, not all cloud inhibition regions have forest loss hotspots like Southeast US. The direct evidence from different cloud data and the indirect evidence from mechanism explanation and the actual impact of forest loss jointly improve the credibility of our results. As we emphasized, more observations at fine-scale are needed to go beyond the global pattern shown and dive into regional and specific cloud impact.

In the revision, we have a dedicated paragraph to discuss the robustness, uncertainty of the estimated cloud effects, as well as the comparison with other studies:

It is worth noting that the estimated cloud effects, despite their broad agreement in the global pattern across datasets, can differ in magnitude and even in sign in certain regions (e.g., the inconsistent cloud effects and their seasonality in central Africa), suggesting cloud data are a key uncertainty source for our analysis. The cloud effects of forests estimated in a recent study⁵⁷ based on a different MODIS-derived cloud dataset⁵⁸ (produced by a different retrieval algorithm) revealed a similar global dominance of cloud enhancement and regional prevalence of cloud inhibition (e.g., in Amazon). Yet regional inconsistencies remained for Southeast US and Central Africa, where the identified cloud inhibition could be weakened or absent with alternative cloud data. This emphasizes the need of using multi-source cloud data to improve the robustness of the estimated cloud effects while reducing the uncertainty from data. Nevertheless, the occurrence of cloud inhibition in these regions is indirectly supported by lower sensible heat over forests, which is in line with the mesoscale circulation mechanism, as well as the consistency between potential cloud inhibition and the actual cloud increase over forest loss locations (i.e., Southeast US hotspot). While more direct observational evidence is always desirable to help resolve inconsistencies, the lack of observations in regions like Central Africa, which have received less attention than the Amazon, hinders comparisons against other available evidence. This highlights the importance of dedicated observational efforts in specific regions, especially those understudied, to provide complementary information to our global-scale analysis.

References

1. Duveiller, G., Hooker, J. & Cescatti, A. The mark of vegetation change on Earth's surface energy balance. *Nat. Commun.* **9**, (2018).
2. Duveiller, G. *et al.* Revealing the widespread potential of forests to increase low level cloud cover. *Nat. Commun.* **12**, 4337 (2021).
3. Li, Y. *et al.* Local cooling and warming effects of forests based on satellite observations. *Nat. Commun.* **6**, 6603 (2015).
4. Khanna, J., Medvigy, D., Fueglistaler, S. & Walko, R. Regional dry-season climate changes due to three decades of Amazonian deforestation. *Nat. Clim. Chang.* **7**, 200–204 (2017).
5. Lawrence, D. & Vandecar, K. Effects of tropical deforestation on climate and

agriculture. *Nat. Clim. Chang.* **5**, 27–36 (2014).

6. Teuling, A. J. *et al.* Observational evidence for cloud cover enhancement over western European forests. *Nat. Commun.* **8**, 14065 (2017).

7. Spracklen, D. V. & Garcia-Carreras, L. The impact of Amazonian deforestation on Amazon basin rainfall. *Geophys. Res. Lett.* **42**, 9546–9552 (2015).

8. Stengel, M. *et al.* Cloud property datasets retrieved from AVHRR, MODIS, AATSR and MERIS in the framework of the Cloud-CCI project. *Earth Syst. Sci. Data* **9**, 881–904 (2017).

REVIEWERS' COMMENTS

Reviewer #1 (Remarks to the Author):

In their revised manuscript "Contrasting impacts of forests on cloud cover based on satellite observations", the authors evaluate how changes in forest cover correlate with observed changes in cloud cover using satellite data. In the last round of review, I asked that the authors compare their results to those of a similar recent study (Duveiller 2021 / D2021); in their response, the authors systematically address the methodological differences between the two studies, finding that the largest differences between the two studies can be attributed to differences in the cloud datasets (which both came from MODIS, but were post-processed differently) and differences in methodology regarding what is considered a change in forest cover. I commend the authors on their rigorous comparison to D2021, in particular for applying the methodology from this paper to the data from D2021. The authors add a condensed discussion of the differences between their study and D2021 to the revised manuscript.

I recommend Nature Communications accept this manuscript for publication, following the data used being made publicly available following the FAIR data principles (the authors provide a Figshare address, but the url does not work for me, nor does searching the doi on Figshare).

Reviewer #2 (Remarks to the Author):

I would like to thank the authors for their response and the additional work in comparing D2021 data and methodology to their work. I also appreciate the fact that the updated text now better acknowledges the uncertainties between sign of the effect that was previously only shown in a supplemental figure.

I have two minor comments that could be included into the manuscript before publication.

1. When discussing the role of uncertainty in the updated text, it may make sense to reference the supplemental figure 13 in addition to D2021.
2. I would also suggest to specifically state the main methodological difference between your work and D2021 in the updated text.

REVIEWERS' COMMENTS

Reviewer #1 (Remarks to the Author):

In their revised manuscript “Contrasting impacts of forests on cloud cover based on satellite observations”, the authors evaluate how changes in forest cover correlate with observed changes in cloud cover using satellite data. In the last round of review, I asked that the authors compare their results to those of a similar recent study (Duveiller 2021 / D2021); in their response, the authors systematically address the methodological differences between the two studies, finding that the largest differences between the two studies can be attributed to differences in the cloud datasets (which both came from MODIS, but were post-processed differently) and differences in methodology regarding what is considered a change in forest cover. I commend the authors on their rigorous comparison to D2021, in particular for applying the methodology from this paper to the data from D2021. The authors add a condensed discussion of the differences between their study and D2021 to the revised manuscript.

I recommend Nature Communications accept this manuscript for publication, following the data used being made publicly available following the FAIR data principles (the authors provide a Figshare address, but the url does not work for me, nor does searching the doi on Figshare).

We are grateful for recommending acceptance of our work. We are sorry that the data doi was not working. We tested and found no issues on our side. The doi (10.6084/m9.figshare.15081510) can be accessed with the following address:

<http://doi.org/10.6084/m9.figshare.15081510>

Reviewer #2 (Remarks to the Author):

I would like to thank the authors for their response and the additional work in comparing D2021 data and methodology to their work. I also appreciate the fact that the updated text now better acknowledges the uncertainties between sign of the effect that was previously only shown in a supplemental figure.

I have two minor comments that could be included into the manuscript before publication.

1. When discussing the role of uncertainty in the updated text, it may make sense to reference the supplemental figure 13 in addition to D2021.

Thanks. We added reference to Figure S13 in Line 329:

Yet regional inconsistencies remained for Southeast US and Central Africa, where the

identified cloud inhibition could be weakened or absent with alternative cloud data (Ref⁵⁷ and Supplementary Fig. 13).

2. I would also suggest to specifically state the main methodological difference between your work and D2021 in the updated text.

Thanks for your suggestion. We added a summary of the main methodological difference with D2021 in Line 430:

Unlike using direct comparison in cloud cover (and other biophysical variables) between forest and nonforest, an alternative method is to utilize the regression coefficients of cloud cover (dependent variable) to land cover fraction (independent variable) and estimate cloud effects assuming 100% land conversion, as adopted by ref⁵⁷. The alternative regression-based approach is mathematically more complicated, and its implementation involves non-trivial post-processing compared with our method while producing qualitatively similar results.